# UltraGauss: Ultrafast Gaussian Reconstruction of 3D Ultrasound Volumes

**Mark C. Eid** [1,2,✉]  **Ana I.L. Namburete**[2]  **João F. Henriques**[1]

[1] Visual Geometry Group (VGG), University of Oxford, U.K.
[2] Oxford Machine Learning in NeuroImaging Lab (OMNI), University of Oxford, U.K.
✉ markeid@robots.ox.ac.uk

## Abstract

Ultrasound imaging is widely used due to its safety, affordability, and real-time capabilities, but its 2D interpretation is highly operator-dependent, leading to variability and increased cognitive demand. We present **UltraGauss**: an ultrasound-specific Gaussian Splatting framework that serves as an efficient approximation to acoustic image formation. Unlike projection-based splatting, UltraGauss renders by *probe-plane intersection* with in-plane aggregation, aligning with plane-based echo sampling while remaining fast and memory-efficient. A stable parameterisation and compute-aware GPU rasterisation make this method practical at scale. On clinical datasets, UltraGauss delivers state-of-the-art 2D-to-3D reconstructions in minutes on a single GPU (reaching 0.99 SSIM within ∼20 minutes), and a clinical expert survey rates its reconstructions the most realistic among competing methods. To our knowledge, this is the first Gaussian Splatting approach tailored to ultrasound 2D-to-3D reconstruction. Our code is available at: *https://www.robots.ox.ac.uk/~vgg/research/UltraGauss/*

## 1 Introduction

Ultrasound (US) is a mainstay in medical imaging: it is real-time, low-cost, portable, and non-ionising. Yet routine use still asks clinicians to infer 3D anatomy from 2D slices: a cognitively demanding step that introduces operator-dependent variability (Benacerraf, 2002; Nelson & Pretorius, 1998), undermining reproducibility and hindering standardised assessment. While volumetric (3D) probes exist, their workflows are largely offline and the hardware is costly, making them uncommon outside well-resourced centres (Merz & Welter, 2005). Enabling 3D reasoning from routine 2D acquisitions offers a software-only path to scale volumetric assessment across sites and resource levels. As 2D is the universal denominator in global healthcare, 2D-to-3D reconstruction will yield standardised volume and surface metrics that are difficult to obtain consistently from single slices.

Recent learning-based approaches span *implicit* NeRF-style representations (Eid et al., 2025; Yeung et al., 2024; Gaits et al., 2024; Wysocki et al., 2023) and *explicit* voxel grids (Solberg et al., 2007). However, implicit fields are computationally heavy; voxel grids incur memory and resolution limits; and many methods adopt light-transport assumptions (ray accumulation with transmittance), which fundamentally mismatch US physics, where waves propagate into tissue and return to the probe (Powles et al., 2018; Aldrich, 2007). Classical Gaussian Splatting (GS) for cameras achieves fast, high-quality rendering in optical settings by projecting 3D Gaussians onto the image plane and blending splats in depth order using alpha compositing (Kerbl et al., 2023). US images, on the other hand, are not perspective renderings: they sample echo intensities within the probe plane (with attenuation), so the camera-style projection and occlusion paradigm does not apply.

We introduce **UltraGauss**: an ultrasound-specific GS reconstruction framework that serves as an efficient approximation to the US image formation model, replacing projection-based rendering with probe-plane intersection rendering, consistent with wave-based acquisition. Instead of marching rays or *projecting* Gaussians into 2D, UltraGauss evaluates anisotropic 3D Gaussians where they *intersect* the probe plane and aggregates intensities in-plane. This removes the need for depth-based occlusion, matches the acquisition geometry of linear *and* curvilinear probes, and enables resolution-free slicing at arbitrary orientations. This approximation preserves plane-based sampling and dominant attenuation behaviour while avoiding expensive wave simulation.

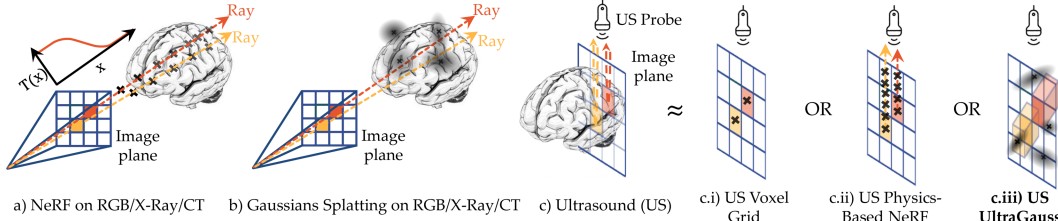

a) NeRF on RGB/X-Ray/CT    b) Gaussians Splatting on RGB/X-Ray/CT    c) Ultrasound (US)    c.i) US Voxel Grid    c.ii) US Physics-Based NeRF    **c.iii) US UltraGauss**

Figure 1: Image formation mechanisms. (*a, b*) Projection-based (e.g. camera) rendering with depth-ordered compositing and (*c*) ultrasound. (*c.i*) Slice sampling from a voxel grid or continuous field. (*c.ii*) Physics-based ultrasound: path integration with attenuation/backscatter. (*c.iii*) **UltraGauss**: plane-intersection of 3D Gaussians with in-plane aggregation.

Our technical contributions are:

• **Efficient forward-model approximation:** UltraGauss renders by *probe-plane* intersection rather than camera projection, capturing plane-based sampling and attenuation while side-stepping costly wave simulation.

• **Stable parameterisation:** A *triangular inverse-covariance (precision)* factorisation guarantees positive-definiteness and well-conditioned gradients for reconstruction and densification.

• **Compute-aware rasterisation:** *Closed-form $\chi^2$ ellipsoidal bounds* in probe coordinates yield tight per-plane 2D boxes, coupled with a *two-phase, load-balanced CUDA* pipeline that rejects non-intersecting Gaussians and confines work to those boxes.

• **Lightweight attenuation:** A Beer–Lambert term models acoustic shadowing when probe geometry is available, adding realism at negligible cost.

• **Clinical validation:** Experiments on clinical fetal datasets and freehand cinesweeps show high-fidelity 3D reconstructions (strong SSIM) within minutes on a single GPU, with clinician-perceived realism comparable to native 3D scans.

## 2 RELATED WORK

**2D-to-3D reconstruction in medical imaging.** Classical pipelines segment 2D slices, register them (e.g. using B-splines), and interpolate a 3D volume (Sarmah et al., 2023). These approaches require substantial manual intervention, struggle with irregular sampling, and scale poorly to high-resolution data. Structure-from-Motion (SfM) methods (e.g. COLMAP (Schönberger & Frahm, 2016)) triangulate depth from photometrically consistent views, but ultrasound violates these assumptions and precise multi-view registration is often impractical.

**Learned volumetric representations.** In US imaging, three requirements shape learned volumetric methods: coping with irregular plane sampling, avoiding external probe tracking, and adopting a rendering model aligned with plane-based echo acquisition (Fig. 1c) rather than camera projection (Fig. 1a-b). Yeung et al. (2024) (ImplicitVol) reconstructs volumes from routine sweeps without IMU/vision tracking, and Eid et al. (2025) (RapidVol) introduces a tensor-factorised hybrid that accelerates training. These advances improve data efficiency and robustness, yet the underlying rendering remains tied to point/voxel sampling using neural networks.

**Ultrasound-specific forward models.** Physics-based renderers estimate position-dependent acoustic parameters (e.g. attenuation, scattering) and synthesise images along wave paths (Wysocki et al., 2023; Guo et al., 2024b) (Fig. 1c.ii). These models can be accurate with known probe poses and source geometry, but they necessitate precise acquisition; curvilinear probes (common in obstetrics) further require cone-to-Cartesian transforms, increasing complexity and sensitivity to error.

**GS in medical imaging.** GS replaces implicit fields with explicit 3D Gaussians and differentiable rasterisation, enabling fast, high-quality rendering. Medical adaptations have largely targeted projection-based modalities: endoscopy (RGB), MRI, and CT (Guo et al., 2024a; Liu et al., 2024; Xie et al., 2024; Bonilla et al., 2024; Peng et al., 2025; Zha et al., 2024; Cai et al., 2025; Nikolakakis et al., 2024). These, therefore, retain camera-style projection and depth compositing (Fig. 1b). Zha et al. (2024) introduce a CUDA-based Gaussian voxel former, but only as a regulariser at small spatial resolutions, and plane sampling from voxels reintroduces interpolation artefacts.

**Ultrasound-adapted GS.** As far as we know, no ultrasound-adapted GS method exists for 2D-to-3D reconstruction. UltraGauss extends GS to volumetric US by replacing projection-based rendering with probe-plane intersection rendering (Fig. 1c.iii): an efficient approximation to the US

image formation model that aligns with plane-based echo sampling and attenuation. The continuous (non-voxelised) Gaussian representation supports arbitrary resolution and orientation slicing without external probe tracking, yielding fast, memory-efficient, and clinically relevant reconstructions.

## 3 BACKGROUND

For cameras in the visible spectrum, one can render the color $c_{\text{RGB}}$ for a given pixel by evaluating a volumetric model along the corresponding ray, sampling the model's opacities $\hat{\alpha}(x)$ and colors $\hat{c}(x)$ at $m$ ordered sample points $x_j$ on the ray:

$$c_{\text{RGB}} = \sum_{j=1}^{m} T_j \hat{\alpha}\left(x_j\right) \hat{c}\left(x_j\right), \quad T_j = \prod_{k=1}^{j-1} \left(1 - \hat{\alpha}\left(x_k\right)\right), \tag{1}$$

where $T_j$ denotes the accumulated transmittance of the material. The main purpose of the accumulated transmittance is to model occlusions (see Fig. 1a – points at the back are blocked by those in front, so are given a lower $T_j$/importance). In the case of GS (Kerbl et al., 2023), the volumetric model is a weighted sum of $n$ Gaussian functions with means $\mu_i$, covariances $\Sigma_i$, colors $c_i$, and coefficients (maximum opacities) $\alpha_i$. In contrast to NeRFs (Mildenhall et al., 2020), the sampled points $x_j$ can now be reduced to the projections of close Gaussians onto the image plane, which can be made more efficient. The opacity of the $i^{\text{th}}$ Gaussian at a 2D (image-space) point $x$ is then calculated as

$$\hat{\alpha}_i\left(x\right) = \alpha_i \exp\left(-\frac{1}{2} \underbrace{\left(x - \mu_i^{\text{2D}}\right)^T \left(\Sigma_i^{\text{2D}}\right)^{-1} \left(x - \mu_i^{\text{2D}}\right)}_{\text{2D squared Mahalanobis distance}}\right), \tag{2}$$

with the mean $\mu_i^{\text{2D}}$ and covariance $\Sigma_i^{\text{2D}}$ projected to the 2D image. To achieve this, the 3D parameters $(\mu_i, \Sigma_i)$ are translated and rotated to the camera's reference frame by a view transformation (extrinsic camera matrix) $W$, and projected to 2D with the affine approximation of a projective transformation (using intrinsic camera matrix $K$) (Zwicker et al., 2001). We can express this formally with operators to convert to and from homogenous coordinates, $\hbar(u) = [u_1, \ldots, u_D, 1]^T$ for $u \in \mathbb{R}^D$, and $\hbar^{-1}(u') = \left[u_1'/u_{D+1}', \ldots, u_D'/u_{D+1}'\right]^T$ for $u' \in \mathbb{R}^{D+1}$ respectively:

$$\mu_i^{\text{2D}} = \text{proj}\left(\mu_i\right) = \hbar^{-1}\left(KW\hbar\left(\mu_i\right)\right) \tag{3}$$

$$\Sigma_i^{\text{2D}} = J_i W \Sigma_i W^T J_i^T, \quad J_i = \frac{\partial \text{proj}(\mu_i)}{\partial \mu_i} \tag{4}$$

One can then combine the opacities and colors of all close Gaussians to obtain the rendered opacity $\hat{\alpha}\left(x\right)$ and color $\hat{c}\left(x\right)$ for use in Eq. 1:

$$\hat{\alpha}\left(x\right) = \sum_i^n \hat{\alpha}_i\left(x\right), \quad \hat{c}\left(x\right) = \frac{1}{\hat{\alpha}\left(x\right)}\left(\sum_i^n \hat{\alpha}_i\left(x\right) c_i\right) \tag{5}$$

In this summary we leave out spherical harmonics, which support directionally-dependent colors (Ramamoorthi, 2006; Yu et al., 2021).

### 3.1 OPTIMIZATION

While Eqs. 1-5 can render a Gaussian model, practical use requires optimizations to avoid costly nested iterations. Namely, Gaussians are tiled and depth-sorted, then distributed across GPU threads for rasterization onto pixels (Kerbl et al., 2023). Depth sorting and cut-off distances for Gaussians in the image plane make rendering only approximate (Huang et al., 2024). Heuristics are necessary to remove/resample Gaussians in overly sparse/dense regions (Rota Bulò et al., 2025; Yu et al., 2023).

## 4 METHOD

### 4.1 IMAGE FORMATION MODEL – ULTRASOUND VS. VISUAL SPECTRUM

While fast splatting is well-developed for RGB cameras, for ultrasound it requires rethinking many design choices due to differences in image formation. US probes use reflected ultrasound waves to measure the response of materials at a dense range of depths from the probe (Fig. 1c). Image

cameras, in contrast, measure light that typically reflects off solid objects, and thus are more affected by occlusions and most often measure surfaces (Fig. 1a-b). Accumulated transmittance $T_j$ in Eq. 1 is therefore unsuitable, as it would treat opaque volumes as occlusions (see $T(x)$ subplot in Fig. 1a). Instead, the key mechanism for ultrasound is detecting *intersections* with the probe plane (Fig. 1c), not projections, while accumulated transmittance plays only a secondary role (discussed in Sec. 4.5).

We do this by first opting to include an additional uniform component (background) with color $c_{\mathrm{BG}}$ and coefficient $\alpha_{\mathrm{BG}}$, which improves numerical stability by avoiding a division by zero. By reusing the definition of $\hat{c}(x)$ from Eq. 5 to combine the gaussian and background components at a 2D point (pixel) $x$, our rendering equation simply becomes:

$$\hat{\alpha}(x) = \sum_i^n \hat{\alpha}_i(x) + \alpha_{\mathrm{BG}} \tag{6}$$

$$c_{\mathrm{Ultrasound}}(x) = \hat{c}(x) = \frac{1}{\hat{\alpha}(x)} \left( \sum_i^n \hat{\alpha}_i(x) c_i + \alpha_{\mathrm{BG}} c_{\mathrm{BG}} \right) \tag{7}$$

A large conceptual difference from rendering the visual spectrum is that, instead of projecting the Gaussian parameters to 2D, we need them to *intersect*, or touch, the probe plane in 3D. The opacity $\hat{\alpha}_i(x)$ of the $i^{\mathrm{th}}$ Gaussian at a 2D point $x$ (necessary for Eqs. 6-7) is then evaluated as a Mahalanobis distance in 3D space, by *lifting the 2D image point to 3D* in the coordinate-frame of the probe:

$$\hat{\alpha}_i(x) = \alpha_i \exp\left( -\frac{1}{2} \underbrace{\left(x_{|0} - \mu_i^{\mathrm{3D}}\right)^T \left(\Sigma_i^{\mathrm{3D}}\right)^{-1} \left(x_{|0} - \mu_i^{\mathrm{3D}}\right)}_{\text{3D squared Mahalanobis distance}} \right), \tag{8}$$

where $x_{|0} = [x_1, x_2, 0]^T$ and the Gaussian's parameters are moved to the probe's coordinate-frame using its inverse transform matrix $W$:

$$\mu_i^{\mathrm{3D}} = \hbar^{-1}\left(W \hbar(\mu_i)\right), \quad \Sigma_i^{\mathrm{3D}} = W \Sigma_i W^T. \tag{9}$$

Contrast Eq. 2 to Eq. 8: despite the similarities, the former (for RGB images) evaluates a 2D Gaussian after projecting it to image-space, while the later evaluates a 3D Gaussian by doing the reverse operation (for ultrasound images). Using this model, we can now design a fast splatting strategy to avoid the computational expense of simply evaluating Eqs. 6-7 and Eq. 8 (which would result in nested iterations over all pixels and all Gaussians).

## 4.2 TRIANGULAR COVARIANCE PARAMETERISATION FOR EFFICIENT INVERSION AND GAUSSIAN SAMPLING

One challenge in optimizing representations with covariance matrices $\Sigma$ (omitting the subscript $i$ for conciseness) is that they must remain positive-definite (PD, all eigenvalues strictly positive), while gradient-based optimization methods typically only support unconstrained optimization. Kerbl et al. (2023) achieved this by reparameterising the covariances as a product of a scaling vector $s$ and a quaternion-derived rotation matrix $R$, as $\Sigma = R\,\mathrm{diag}\left(s^2\right) R^T$. When a Gaussian is projected onto a 2D plane, its covariance becomes 2D, which is easily invertible for use in Eq. 2. However, we found that for our setting the inversion of $\Sigma^{\mathrm{3D}}$ (as opposed to $\Sigma^{\mathrm{2D}}$) when it is formed by a quaternion (which to represent a 3D rotation must first be normalized), resulted in numerical instabilities. Being a $3 \times 3$ matrix, inversion also takes longer. Hence, we propose to learn $\left(\Sigma^{\mathrm{3D}}\right)^{-1}$ directly. We ensure it is PD by parameterising it as a product of a matrix $M$ with itself (which is positive-semidefinite), and adding a small multiple of the identity $I$ to ensure positive eigenvalues (with $\beta > 0$):

$$\Sigma'^{-1} = MM^T + \beta I, \quad M = \begin{bmatrix} M_{11} & M_{12} & M_{13} \\ M_{12} & M_{22} & M_{23} \\ M_{13} & M_{23} & M_{33} \end{bmatrix} \tag{10}$$

Note that $M$ itself is symmetric, to achieve the minimal number of degrees of freedom of a 3D covariance (6). No normalization is needed. A remaining challenge is that, in addition to requiring the inverse covariance $\Sigma^{-1}$ frequently to render pixels (Eq. 8), we also occasionally (e.g. every 100 iterations) must perform heuristic resampling of some Gaussians. This operation requires *inverting* $\Sigma^{-1}$ *explicitly* to obtain the original covariance $\Sigma$, as well as factorizing it to draw a sample from the multivariate Gaussian (Sec. 3.1). Both operations can be numerically unstable for ill-conditioned

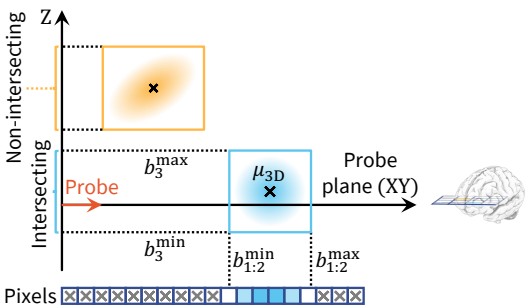

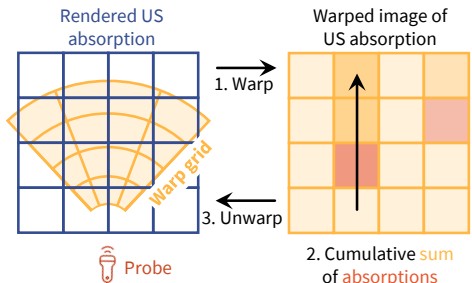

Figure 2: Side view of the Gaussian bounding-box/plane intersection used to derive 2D rasterisation bounds (Sec. 4.3). Boxes that do *not* intersect the probe plane are rejected early (Sec. 4.4); intersecting boxes are rasterized only over their 2D intersection window (Sec. 4.3). Pixels not iterated over are marked with ×. The probe plane (XY) points into the page.

Figure 3: Approximating acoustic shadowing via *warp–integrate–unwarp*. Left: rendered ultrasound absorption in image space. Right: the same field warped to a beam-aligned grid. (1) Warp to beam space; (2) take a cumulative sum along the beam direction (path integral of absorption); (3) unwarp back to image space to obtain the attenuation map.

$\Sigma^{-1}$. Therefore, we propose instead a more efficient parameterisation, as a product of a lower-triangular matrix $L$:

$$\Sigma^{-1} = LL^T, \quad L = \begin{bmatrix} L_{11}^2 + \beta & 0 & 0 \\ L_{12} & L_{22}^2 + \beta & 0 \\ L_{13} & L_{23} & L_{33}^2 + \beta \end{bmatrix}. \tag{11}$$

Eq. 11 guarantees that $\Sigma^{-1}$ is PD, since a triangular matrix's eigenvalues are its diagonal elements, and it can be seen that these are strictly positive (with $\beta > 0$). Moreover, a lower-triangular matrix is extremely efficient to invert via *forward substitution*, which is implemented in most numerical packages. This allows easily computing $\Sigma = \left(L^{-1}\right)^T L^{-1}$. Finally, another advantage is that this factorization allows sampling from a Gaussian (for the resampling heuristic, Sec. 3.1), by simply projecting a standard normal sample $z$ with the (efficiently) inverted $L$:

$$y = \mu + L^{-T}z, \quad z \sim \mathcal{N}(0,1). \tag{12}$$

Eq. 11 supports efficient calculation of $\Sigma^{-1}$, and thus opacity $\hat{\alpha}_i(x)$ (Eq. 8), $1.40\times$ faster than had we used $\Sigma = R\,\mathrm{diag}\left(s^2\right)R^T$. Similarly $\Sigma$ and the resampling heuristics are computed $1.25\times$ faster (see Appendix A for more details). Yet for most Pixel-Gaussian pairs, opacity is near 0 and so can be ignored. Sec. 4.3 discusses this.

## 4.3 RASTERIZATION BOUNDARIES

To avoid evaluating Eq. 8 when it is known to yield opacities close to 0, we compute it only inside a bounding box for each Gaussian. The bounding box is defined around the ellipsoid that encompasses $p\%$ of the Gaussian's probability density. This ellipsoid can be expressed as a function of the 3D squared Mahalanobis distance from Eq. 8, and the chi-squared distribution value for 3 degrees of freedom (omitting the subscript $i$ for clarity):

$$\left(x_{|0} - \mu^{3D}\right)^T \left(\Sigma^{3D}\right)^{-1} \left(x_{|0} - \mu^{3D}\right) \leq \chi_{3,1-p}^2, \tag{13}$$

which evaluates to $\chi_{3,1-p}^2 = 7.815$ for $p = 95\%$ (note that it is a bound on a squared distance; see Appendix G.3 for ablation on $p$ value). To calculate this ellipsoid's bounding box, its bounds are the 3D vectors $b^{\min}$ and $b^{\max}$:

$$b^{\min/\max} = \mu^{3D} \pm \sqrt{\lambda v}, \quad \lambda = \frac{\chi_{3,1-p}^2}{\det\left(\Sigma^{3D}\right)}, \tag{14}$$

$$v = \begin{bmatrix} \Sigma_{22}^{3D}\Sigma_{33}^{3D} - \left(\Sigma_{23}^{3D}\right)^2 \\ \Sigma_{11}^{3D}\Sigma_{33}^{3D} - \left(\Sigma_{13}^{3D}\right)^2 \\ \Sigma_{11}^{3D}\Sigma_{22}^{3D} - \left(\Sigma_{12}^{3D}\right)^2 \end{bmatrix},$$

using $\Sigma_{jk}^{3D}$ to index the $(j,k)^{\text{th}}$ element of a matrix. Eq. 14 can be obtained by writing Eq. 13 as a function of each element of $\delta = x_{|0} - \mu^{3D}$. Fig. 2 illustrates these bounds.

### 4.4 LOAD BALANCING ACROSS RENDERING THREADS

Note that the bounding box (Eq. 14) can be partitioned into 2 components:

1. The 2D bounding box in the probe plane (first two elements of $b^{\min/\max}$, i.e. $b_{1:2}^{\min/\max}$).

2. The 1D segment orthogonal to the probe plane (the third element, $b_3^{\min/\max}$).

This suggests a two-phase process for efficient rendering:

1. Reject any Gaussians whose cut-off boundaries do not intersect with the probe plane: $b_3^{\min} > 0$ or $b_3^{\max} < 0$.

2. For each accepted Gaussian, only iterate over 2D pixels $x$ inside the bounding box of the plane: $b_1^{\min} \leq x_1 \leq b_1^{\max}$ and $b_2^{\min} \leq x_2 \leq b_2^{\max}$.

This process is shown in Fig. 2, and naturally balances the load across parallel GPU threads. Phase 1 requires iterating through all the Gaussians (but not the pixels), marking them as accepted or rejected based on the perpendicular distance, which can be equally partitioned between all the threads. Standard buffer compaction (Corp., 2020) can then reduce this list of Gaussians to only the accepted ones. Phase 2 then requires iterating through the compacted list of only accepted Gaussians, and rasterizing each one only onto the corresponding bounding box of the image buffer, by atomically adding to pixel accumulators for color $\hat{c}(x)$ and opacity $\hat{\alpha}(x)$ (implementing Eqs. 6-7). The compacted list in Phase 2 is again equally partitioned among the threads, to ensure optimal throughput.

### 4.5 SHADOW MODELLING

While other NeRF-based methods model more complex ultrasound physics (Guo et al., 2024b; Wysocki et al., 2023), we found that a simple and fast approximation can account for the attenuation of ultrasound waves as they traverse from the probe to a point $x$, producing shadows along the way. Based on the Beer-Lambert Law, we calculate an intensity reduction factor $T$ for a pixel located at position $x$:

$$T(x) = \exp^{\int_{s=0}^{s=x} \alpha(s) ds} \approx \exp^{-\sum_{j=0}^{j=N} \hat{\alpha}_j \delta_j} \tag{15}$$

where $\hat{\alpha}_j$ is the opacity of pixel $j$ as computed in Eq. 6, and $\delta_j$ is the distance between successive $\hat{\alpha}_j$'s, which in our case is simply the pixel spacing ($= 1/\text{ImageHeight}$). This can be efficiently computed for all pixels by using a cumulative sum (`cumsum`) along the row dimension, after warping (bilinearly interpolating) the ultrasound cone into a square image (illustrated in Fig. 3). We then multiply $c_{\text{Ultrasound}}(x)$ from Eq. 7 by $T(x)$ to give the updated image pixel color.

### 4.6 OPTIMIZATION

We train UltraGauss end-to-end by backpropagation with the Adam optimizer, optimizing per-Gaussian parameters $\{\mu_i, L_i, c_i, \alpha_i\}$. From our triangular parameterisation, precision and covariance are linked by

$$\Lambda_i \equiv \Sigma_i^{-1} = L_i L_i^\top + \varepsilon I \quad \text{(cf. Eq. 11)},$$

so we optimize $L_i$ (lower triangular) directly and recover $\Sigma_i = \Lambda_i^{-1}$ as needed.

**Initialisation.** Unlike camera GS, we do not use COLMAP initialisation (which yields surface-only points). Instead, we sample $\mu_i$ uniformly within the acquisition volume, and intialise $c_i = 0.5$ and $\alpha_i = 0.731$ (constrained to $[0, 1)$ via a sigmoid). For the precision factor, we draw entries $L_{ij} \sim \mathcal{U}[4, 5]$; under our normalisation this corresponds to small initial marginal variances for $\Sigma_i$ (approximately $(1.7 – 4.3) \times 10^{-3}$). A small jitter $\varepsilon$ ensures $\Lambda_i \succ 0$.

**Training details.** Sparsification and densification heuristics follow (Kerbl et al., 2023; Ye et al., 2024) with small adaptations for plane-intersection rasterisation. We evaluate $N \in \{100k, 200k, 2M\}$ Gaussians; our CUDA kernels sustain $N \approx 2M$ with minute-level reconstructions (fewer Gaussians trade a little accuracy for speed). Learning rates are $0.05$ for all parameters except $\mu_i$, which uses an exponentially decaying schedule starting at $1.6 \times 10^{-4}$ (Yu et al., 2021; Kerbl et al., 2023). All experiments run on a single NVIDIA RTX-A4000. Various ablations can be found in Appendix G. Our code (PyTorch and custom CUDA kernels for forward process and gradients) is available at: *https://www.robots.ox.ac.uk/~vgg/research/UltraGauss/*

## 5 EXPERIMENTS

### 5.1 DATASETS

Two clinical datasets were curated to validate UltraGauss in different ultrasound acquisition settings. **Dataset A** consists of volumetric scans, allowing assessment of reconstruction fidelity across multiple orthogonal views. **Dataset B** comprises freehand US video sequences, evaluating reconstruction quality in the absence of full 3D coverage, mimicking real-world fetal monitoring scenarios.

**Dataset A – 3D Ultrasound Volumes:** This dataset includes twelve 3D fetal brain ultrasound volumes ($160 \times 160 \times 160$ voxels, $0.6 \times 0.6 \times 0.6$ mm$^3$ resolution), obtained from the INTERGROWTH-21$^{st}$ study (Papageorghiou et al., 2018). Acquisitions were performed between 14 and 26 gestational weeks: spanning a critical period of brain maturation (Namburete et al., 2015; 2023), and the standard time for fetal anomaly screening (Salomon et al., 2022). The scans were collected using a Philips HD9 curvilinear probe (2.5 MHz wave frequency) by multiple sonographers, which introduces variability in probe positioning and image appearance.

**Dataset B – 2D Freehand Video Sequences:** Three freehand 2D ultrasound videos of fetal brain acquisitions were collected at 19 and 20 weeks' gestational age at Leiden University Medical Center using a GE Voluson E10 ultrasound scanner. Each video consists of $\sim$100 frames, with each frame cropped and resized to $160 \times 160$ pixels, and resampled to a resolution of $0.6 \times 0.6$ mm$^2$. Fig. D1 (Appendix) illustrates how UltraGauss uses Datasets A and B.

### 5.2 EVALUATION OF RECONSTRUCTION QUALITY AND SPEED

To assess maximum achievable reconstruction quality, we provide UltraGauss with *oracle coverage*: 160 evenly spaced axial slices (sampled from a 3D Scan in Dataset A). This exposes the full $160 \times 160 \times 160$ volume. We evaluate reconstructions by rendering 160 slices in each of the axial, coronal, and sagittal cross-sections, comparing against the native slices using SSIM ($\uparrow$) (Wang et al., 2004), PSNR ($\uparrow$) and LPIPS ($\downarrow$) (Zhang et al., 2018). As reconstruction time $t$ is a key consideration for clinical adoption, we report metrics at $t = 5$ and $t = 20$ minutes, and at convergence ($t = \infty$).

**Limited and perturbed inputs.** We repeat the evaluation with only 50% coverage (80 evenly spaced axial slices), and add small, random rotations about the x and y axes ($\theta \sim \mathcal{U}(-5°, +5°)$) to mimic the sonographer's hand motion and fetal movement encountered in practice.

**Baselines and protocol.** We benchmark against three SOTA *ultrasound* reconstruction models: UltraNerf (Wysocki et al., 2023) (physics-informed ray tracing); RapidVol (Eid et al., 2025) (hybrid implicit-explicit); and ImplicitVol (Yeung et al., 2024) (fully implicit, NeRF-like). To satisfy UltraNerf's parallel-ray assumption, curvilinear inputs are converted from polar to Cartesian.

### 5.3 CLINICIANS' SURVEY OF RECONSTRUCTION QUALITY

We assess perceived clinical realism and fidelity of UltraGauss' reconstructions in a two-part reader study with expert sonographes. All survey material was de-identified.

**Study A: Pairwise realism preference.** Participants compared natively acquired 3D ultrasound images with reconstructions from RapidVol, ImplicitVol, and UltraGauss variants. We evaluated four routinely assessed fetal brain planes at the 20-week anomaly scan: mid-axial, transthalamic, transventricular, and mid-coronal (Salomon et al., 2022). Assessments spanned five models (ImplicitVol, RapidVol, UltraGauss-100K, UltraGauss-300K, UltraGauss-2M), four fetal scans, and three training budgets $t \in \{5, 20, \infty\}$ minutes. Each participant viewed 10 randomized image pairs (order of questions and left/right placement shuffled) and selected the most realistic scan, or indicated "no preference" if indistinguishable. Respondents reported speciality training, years in practice, and confidence in fetal brain assessment.

**Study B: Temporal fidelity ("Turing") test.** To examine convergence over time, a second survey presented nine UltraGauss reconstructions of a mid-coronal plane generated at $t = \{0.5, 1, 2, 3, 4, 5, 10, 15, 20\}$ minutes, alongside two identical ground-truth images. Inputs comprised 80 axial slices, covering only 50% of the 3D volume and orthogonal to the target plane. Participants labelled images as either a real ultrasound scan or an "AI reconstruction". We randomly included duplicated ground-truth images as a control, to quantify variability in expert judgements. The survey given to clinicians can be seen in Appendix C.1.

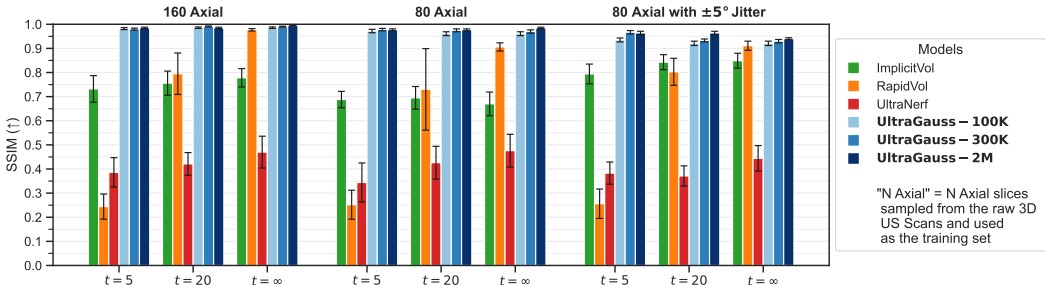

Figure 4: Reconstruction results for 6 different models (ours in **bold**) on 3 different training datasets. Shown at 3 time points over the duration of reconstruction. Higher SSIM is better. Errors bars show $\pm 1$ standard deviation amongst fetuses.

## 5.4 ULTRAGAUSS IN AN END-TO-END CLINICAL PIPELINE

We evaluate UltraGauss in a practical point-of-care workflow: a clinician acquires a freehand video (cinesweep) with a standard sensorless 2D probe, and then reconstructs a full 3D volume for retrospective multiplanar review. This enables imputation of planes that were missed during scanning.

**Pipeline.** Each video frame first receives a 6-DoF pose via an ultrasound pose-estimation model (e.g., Ramesh et al. (2024); Di Vece et al. (2024); Yeung et al. (2022)). Frames and predicted poses are then fed into a suitable 3D reconstruction model (i.e., UltraGauss), or to a baseline (RapidVol, ImplicitVol, UltraNerf).

**Evaluation protocol.** Ground-truth 3D for these fetal cinesweeps is unavailable, so we adopt a frame hold-out cross-validation scheme: for each video, we randomly partition the video frames into 80% training and 20% testing. After reconstruction, we render the held-out slices at their predicted poses and compare the syntheses against the corresponding acquired frames. The residual captures aggregate error from both pose estimation and reconstruction inaccuracies. We run this end-to-end pipeline on all three fetal cinesweeps in Dataset B.

## 5.5 SHADOW MODELLING

We compare UltraGauss with the shadow modelling feature enabled against UltraNerf (Wysocki et al., 2023), using their publicly available Synthetic Liver linear-probe dataset.

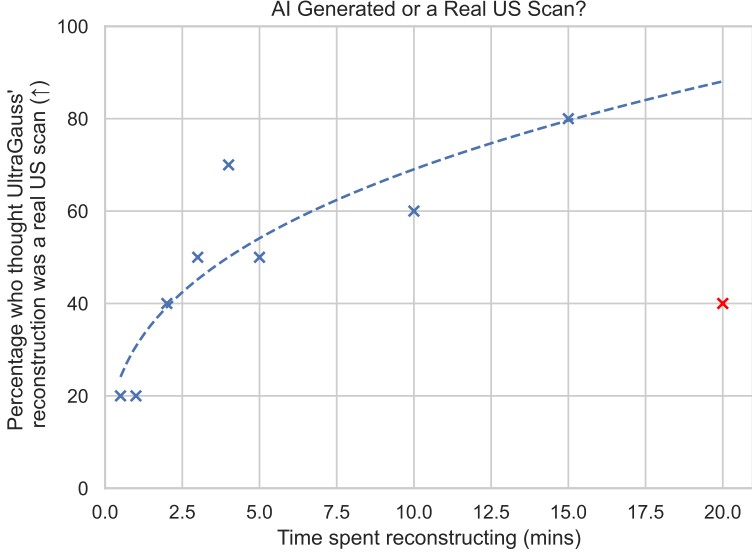

Figure 5: Clinicians' survey results, asking whether an image is real or "AI Generated" (i.e. UltraGauss-generated).

## 6 RESULTS

### 6.1 QUALITY AND SPEED EVALUATION

**Summary.** We evaluate on three test sets of increasing difficulty (Fig. 4, more in Fig. B1). Across all datasets and training budgets $t \in \{5, 20, \infty\}$, *UltraGauss* outperforms RapidVol, ImplicitVol and UltraNerf. It also has high anatomical accuracy and is not organ-specific (Appendices E and F).

**Accuracy vs. time.** For near–real-time use ($t$=5 min), UltraGauss exceeds the best baseline by $\geq 0.20$ SSIM. At convergence, UltraGauss–2M attains a mean SSIM of **0.995**. Variability across gestational ages is markedly lower: at least $10\times$ smaller variance than all baselines.

**Capacity–time trade-offs.** Model capacity interacts with the time budget: (i) at $t$=5 min, $\sim$100k Gaussians perform best; (ii) for long runs ($\mathcal{O}$(hours)), 2M Gaussians yield the highest final accuracy; (iii) $\sim$300k Gaussians offer a strong balance across budgets.

### 6.2 CLINICIANS' SURVEY

**Participants.** We invited 12 expert sonographers specializing in fetal, pediatric, and general ultrasound imaging, from hospitals in four countries (UK, Ghana, Denmark, and the Netherlands). Ten experts responded (consultant fetal surgeons and senior sonographers), with a mean of 18 years' experience (range: 7–30 years).

**Comparison of reconstruction methods:** Under matched short training budgets ($t \leq 20$ minutes), *all* clinicians preferred UltraGauss over RapidVol and ImplicitVol. At convergence ($t$=$\infty$), no clinician rated UltraGauss worse than the alternatives (Appendix C.2). Within UltraGauss, preference increased with capacity (2M > 300K > 100K), indicating that larger Gaussian sets yield more realistic reconstructions.

**Temporal realism.** In the progressive ("Turing") test, where the rendered novel-view was orthogonal to the input data, 70% of clinicians selected UltraGauss as more realistic than the ground-truth after just 4 minutes of training, rising to 80% after 15 minutes (Fig. 5; examples in Fig. B2). As a control for judgement variability, two identical ground-truth images were included; 40% of participants labelled one of these as AI, underscoring the inherent variability in expert assessments.

Table 1: Quantitative results of reconstruction performance on cinesweep videos. The SSIM scores shown are the average across all the frames held-out for testing/used in training, at $t = \infty$. Best scores are highlighted in **bold**. $\uparrow$ indicates higher is better.

|  | Model | Video 1 | Video 2 | Video 3 | Avg. | Std. |
|---|---|---|---|---|---|---|
| Test SSIM $\uparrow$ | ImplicitVol | 0.674 | 0.797 | 0.772 | 0.747 | 0.065 |
|  | RapidVol | 0.745 | 0.799 | 0.760 | 0.768 | 0.028 |
|  | UltraNerf | 0.446 | 0.626 | 0.521 | 0.531 | 0.091 |
|  | **UltraGauss** | **0.928** | **0.905** | **0.910** | **0.914** | **0.012** |
| Train SSIM $\uparrow$ | ImplicitVol | 0.914 | 0.899 | 0.893 | 0.902 | 0.010 |
|  | RapidVol | 0.871 | 0.884 | 0.880 | 0.878 | **0.007** |
|  | UltraNerf | 0.489 | 0.591 | 0.598 | 0.559 | 0.061 |
|  | **UltraGauss** | **0.959** | **0.957** | **0.939** | **0.952** | 0.011 |

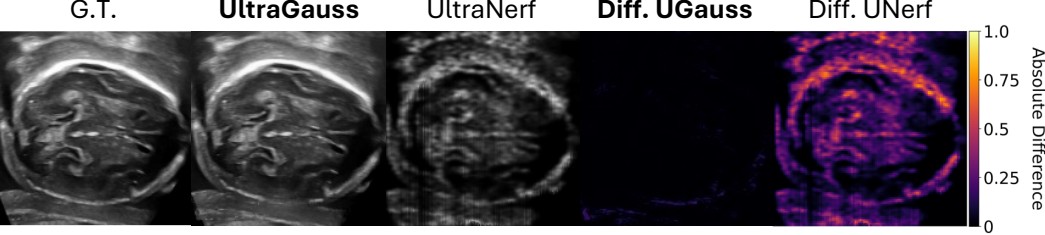

Figure 6: UltraGauss and UltraNerf being used to predict one of the withheld US video frames at its estimated pose. The pixel-wise absolute difference between each prediction and the ground truth image is also shown in the perceptually uniform `inferno` colored heatmap (alongside its color bar). UltraGauss produces a reconstruction that is nearly identical to the Ground Truth (G.T.) Image, as reflected by the near-black absolute difference image ("Diff. UGauss").

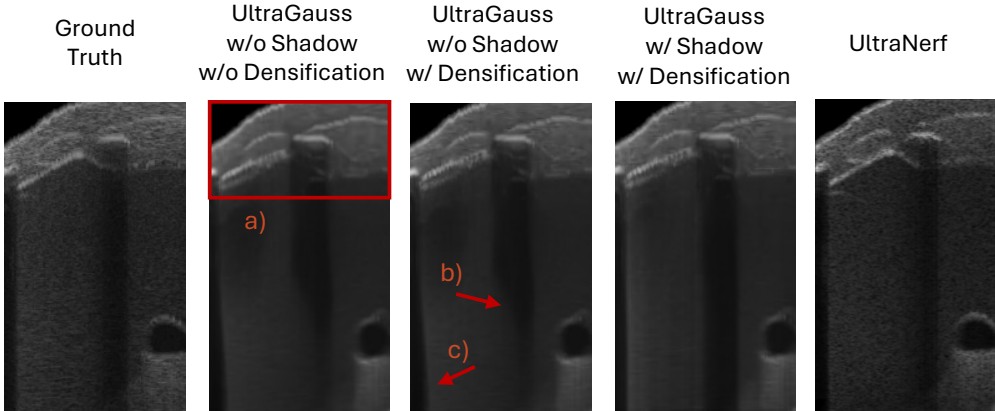

Figure 7: Test-set images rendered with UltraGauss (toggling Gaussian densification or shadow modelling), and UltraNerf. Without densification, speckle is lacking (a). Without shadow modelling, acoustic shadows can abruptly stop (b), or bulge out and become curved instead of straight (c).

### 6.3 END-TO-END PIPELINE RESULTS

We evaluate end-to-end by rendering held-out frames at their *predicted* poses and comparing against the acquired images (Tab. 1; examples in Fig. 6; temporal curves and full visualizations in Appendix D.2). Across videos and training budgets ($t \in \{5, 20, \infty\}$), *UltraGauss* achieves the highest SSIM/PSNR and lowest LPIPS, with especially large gains at short budgets which is relevant for time-critical scans and brief consultations. Averaged over all held-out frames, UltraGauss reaches **SSIM 0.91**. Note that these scores reflect the *combined* error from both pose estimation and reconstruction.

### 6.4 SHADOW MODELLING

Enabling shadow modelling on UltraGauss improves SSIM by 0.005, PSNR by 0.2 dB and LPIPS by 0.004, with practically no time penalty. Whilst the metric gains are minimal, it can be seen in Fig. 7 that it leads to straighter, and more realistic acoustic shadows. More are shown in Fig. G2. Despite these improvements, we still lack some of the speckle arising from multiple-scattering which UltraNerf is able to capture, thanks to their inclusion of scattering density and intensity parameters. However, when the ground truth poses are not known due to acquisition from sensorless 2D probes (Sec. 5.4), or if the ultrasound rays are not perfectly straight (e.g. due to using a curvilinear probe and then transforming the image from polar to Cartesian space, as in Fig. 3), then UltraGauss can yield more robust results, since it is less dependent on a calibrated physics model. This is evident by the results in Secs. 6.1 to 6.3. UltraGauss is also much quicker – $6.94\times$ faster than UltraNerf.

## 7 CONCLUSION

We introduced **UltraGauss**, an ultrasound-specific Gaussian Splatting framework that serves as an *efficient approximation* to acoustic image formation via *probe–plane intersection* rendering. We derived closed-form $\chi^2$ bounds for plane-intersection rasterisation and a two-phase, load-balanced CUDA pipeline, and proposed a triangular inverse-covariance (precision) parameterisation that stabilises optimization at scale. On clinical datasets, UltraGauss delivers minute-level $2D \rightarrow 3D$ reconstructions with high fidelity, and expert sonographers consistently prefer its realism to prior methods. In an end-to-end cinesweep workflow, it enables retrospective multiplanar review from routine 2D acquisitions without additional hardware. By aligning the rendering model with ultrasound physics while retaining the efficiency of Gaussian splatting, UltraGauss provides a practical route to standardised volumetry and more accessible 3D ultrasound, particularly valuable across diverse clinical settings. Future work includes joint pose–reconstruction optimization, richer acoustic effects beyond Beer–Lambert attenuation, and broader evaluation across anatomies and scanners.

### ACKNOWLEDGMENTS

M.E. is kindly supported by the EPSRC CDT in AIMS (EP/S024050/1) and AWS; A.N. by the Bill & Melinda Gates Foundation; J.H. by the Royal Academy of Engineering (RF/201819/18/163).

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

## A  DETAILED ANALYSIS OF OUR COVARIANCE PARAMETERIZATION

### A.1  SPEED IMPROVEMENT

The number of FLOPs (Floating Point Operations) to compute $\left(\Sigma^{3D}\right)^{-1}$, which is required **every** iteration, and $\Sigma^{3D}$, which is only required every $N_{\text{densification}}$ iterations, can be calculated theoretically (Hunger, 2007) as follows:

EXISTING PARAMETERIZATIONS APPLIED TO 3D

1. To form the Rotation matrix $R$ from the 4 quaternions $\{q_r, q_i, q_j, q_k\} \in \mathbb{R}^N$ (where $N$ is the Number of Gaussians), we apply the standard quaternion to rotation matrix conversion and see that it requires $30N$ FLOPs. Forming the diagonal Scaling matrix $S$ from the 3 scaling values $\{s_x, s_y, s_z\} \in \mathbb{R}^N$ requires 0 FLOPs.

2. Next, to form $\Sigma^{3D}$ from $\Sigma^{3D} = R^T S^T S R$, we are first required to form $A = SR$. Since $S$ is diagonal, this only requires $3 \times 3 = 9$ FLOPs. We now have $\Sigma^{3D} = A^T A$ and can make use of the symmetric nature of the product's result, so only have to compute the leading diagonal terms and one of the off-diagonal triangles. This means for a $3 \times 3$ matrix, 30 FLOPs are required to compute this. Thus in total $9 + 30 = 39N$ FLOPs for this step.

3. Finally, to compute $\left(\Sigma^{3D}\right)^{-1}$, we can exploit the properties of the Covariance matrix, namely that it is symmetric and positive definite, and use Cholesky decomposition to invert it in 39 FLOPs. This means that overall, $\Sigma^{3D}$ requires $30N + 0N + 9N + 30N = 69N$ FLOPs, and to form $\left(\Sigma^{3D}\right)^{-1}$, $69N + 39N = 108N$ FLOPs.

OUR PARAMETERIZATION

1. To form

$$
L = \begin{bmatrix} L_{11}^2 + \beta & 0 & 0 \\ L_{12} & L_{22}^2 + \beta & 0 \\ L_{13} & L_{23} & L_{33}^2 + \beta \end{bmatrix} \tag{11}
$$

from our 6 covariance parameters $\{L_{11}, L_{12}, L_{13}, L_{22}, L_{23}, L_{33}\} \in \mathbb{R}^N$ (where $N$ is the Number of Gaussians), it requires $3N$ multiplications and $3N$ additions, so $6N$ FLOPs in total.

2. Next, to form $\left(\Sigma^{3D}\right)^{-1}$ from $\left(\Sigma^{3D}\right)^{-1} = LL^T$, as $L$ is a $M \times M$ Lower Triangular Matrix with $M = 3$, we only require $\frac{1}{3}M^3 + \frac{1}{2}M^2 + \frac{1}{6}M = \frac{1}{3}3^3 + \frac{1}{2}3^2 + \frac{1}{6}3 = 14$ FLOPs. So $14N$ FLOPs in total. Therefore every iteration, we require $6N + 14N = 20N$ FLOPs to compute $\left(\Sigma^{3D}\right)^{-1}$ from our 6 parameters and use it in in Eq. 8 .

3. Finally, to form $\Sigma^{3D}$ from $\Sigma^{3D} = \left(L^{-1}\right)^T L^{-1}$, we first need to invert $L$. However since $L$ is Lower Diagonal, this can be done very efficiently using forward substitution and only takes $\frac{1}{3}M^3 + \frac{2}{3}M = \frac{1}{3}3^3 + \frac{2}{3}3 = 11$ FLOPs. The product of two lower triangular matrices (this time $L^{-1}$ rather than $L$) is, as before, 14 FLOPs. As such, we need $11N + 14N = 25N$ additional FLOPs to form $\Sigma^{3D}$.

Importantly, however, is that during densification we only need $\sigma_{xx}$, $\sigma_{yy}$ & $\sigma_{zz}$, and not the full covariance matrix. This means we can simply take the squared L2-norm of $\left(L^{-1}\right)^T$ along the column dimension, only requiring $9N$ FLOPs. When we then need to sample from the gaussian distributions, we can also make use of Eq. 12, which again does not require $\Sigma^{3D}$ but $\left(L^{-1}\right)^T$ - and that has already been calculated. Therefore, every $N_{\text{densification}}$ iterations we only require $20N$ additional FLOPs.

The above can be nicely summarised in Tab.A1

| Method | FLOPs for $\Sigma^{-1}$ (computed every iter) | FLOPs for $\Sigma$ (computed every $N_{\text{densification}}$ iters) |
|---|---|---|
| Original 3DGS Formulation (e.g., Kerbl et al.; Zha et al.) | $108N$ | $69N$ |
| **UltraGauss (ours)** | $20N$ | $+20N$ in practice$^*$ 
 $(+25N$ if $\Sigma$ is actually needed$^*)$ |

Table A1: Comparison of theoretical FLOPs required to compute $\Sigma$ and $\Sigma^{-1}$ for different parameterizations. $^*$For densification, only the variances are needed - not the full covariance matrix - so we can take the squared L2 norm of $L^{-T}$, saving 5 FLOPs.

As can be seen, our formulation is much more efficient than had we used the 3D version of 3DGS' formulation. Not only do we require $5.4\times$ less FLOPs to compute $\Sigma^{-1}$ each iteration, even if we did need the full covariance matrix $\Sigma$, our formulation still computes this using $1.73\times$ less FLOPs. In practice, we measured speed-ups of $1.4\times$ on the forward pass and $1.2\times$ on the backward pass when calculating $\Sigma^{-1}$. As $\Sigma^{-1}$ is required every iteration, this is a meaningful speed improvement. When $\Sigma$ is required for densification every $N_{\text{Densification}}$ iterations, our formulation computes this $1.25\times$ faster in the forward pass and $6.65\times$ faster in the backward pass than had we used the 3DGS formulation (although in practice we only need $L^{-T}$, not the full covariance matrix, so speed-up is even quicker).

## A.2 NUMERICAL STABILITY

In terms of numerical stability, we also performed a numerical sensitivity analysis on the two formulations. Each input parameter was subjected to a small perturbation, to simulate numerical errors/rounding, and the relative Frobenius Norm difference between the error-containing inverse/covariance matrix and the error-free matrix is shown.

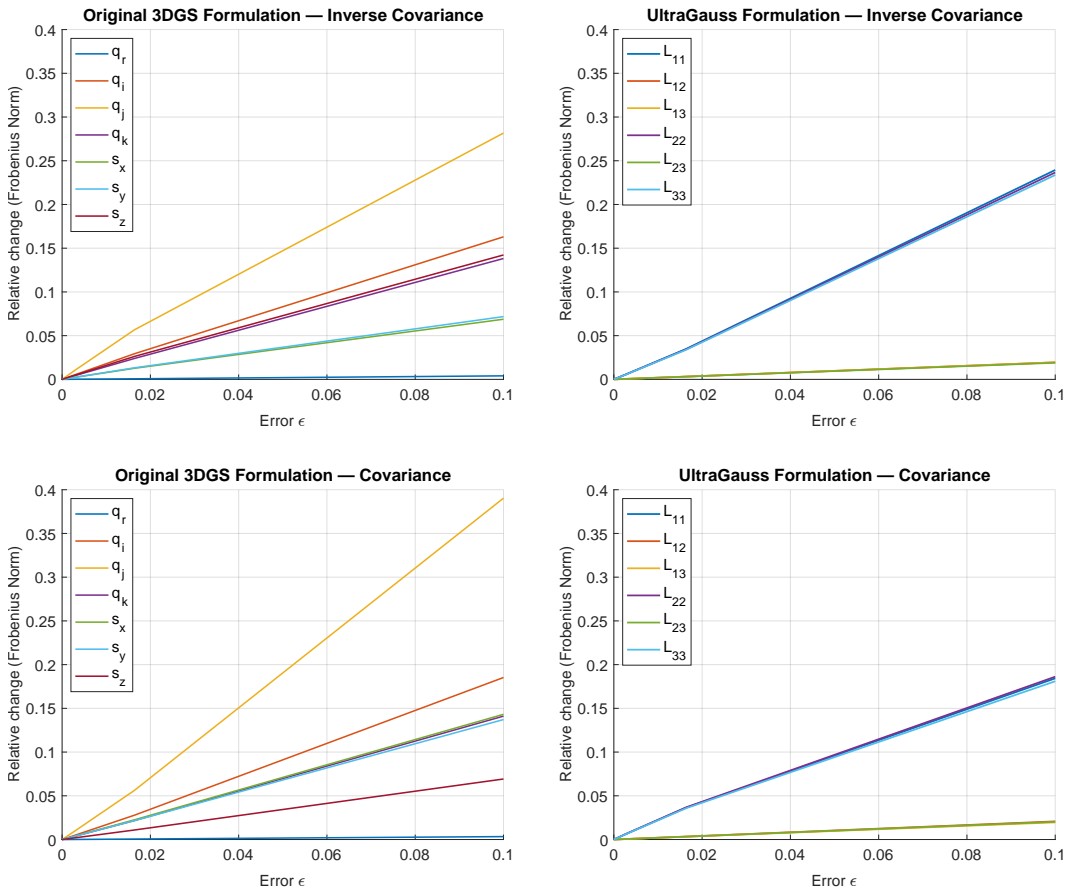

Figure A2: Numerical analysis of the effect a small perturbation or error in each parameter has on the final Covariance or Inverse Covariance Matrix, for the original parameterisation and our proposed one.

From Fig. A2 it is evident that our proposed formulation is more robust to numerical error, when computing both the inverse covariance matrix and covariance matrix from the input parameters.

We also found empirically that sometimes, when trying to invert the $3 \times 3$ covariance matrix formed by the four quaternions and three scaling parameters (which is never done in classical GS, as only the *projected* $2 \times 2$ covariance matrix is inverted), if any of the eigenvalues became close to zero we would get numerical instabilities. Our formulation (by adding a small $\beta$), ensures all eigenvalues are always $\geq \beta$ so we never run into this issue.

### A.3 PHYSICAL MEANING OF $L_{ij}$ TERMS IN EQ. 11

In order to gain an intuitive understanding of the physical meaning of the components of $L$ in Eq. 11, the $3 \times 3$ covariance matrix $\Sigma$ needs to be algebraically calculated, as only then can the standard deviations and correlation coefficients in the $x, y, z$ dimensions be extracted and provide us with geometric insight. Ignoring the small value $\beta$ from Eq. 11 for simplicity:

$$L = \begin{bmatrix} L_{11}^2 & 0 & 0 \\ L_{12} & L_{22}^2 & 0 \\ L_{13} & L_{23} & L_{33}^2 \end{bmatrix} \tag{A1}$$

$$\Sigma^{-1} = LL^T = \begin{bmatrix} L_{11}^4 & L_{11}^2 L_{12} & L_{11}^2 L_{13} \\ L_{11}^2 L_{12} & L_{12}^2 + L_{22}^4 & L_{23}L_{22}^2 + L_{12}L_{13} \\ L_{11}^2 L_{13} & L_{23}L_{22}^2 + L_{12}L_{13} & L_{13}^2 + L_{23}^2 + L_{33}^4 \end{bmatrix} \tag{A2}$$

$$\Sigma = \left[\Sigma^{-1}\right]^{-1} = \begin{bmatrix} \sigma_{xx}^2 & \rho_{xy}\sigma_{xx}\,\sigma_{yy} & \rho_{xz}\sigma_{xx}\,\sigma_{zz} \\ \rho_{xy}\sigma_{xx}\,\sigma_{yy} & \sigma_{yy}^2 & \rho_{yz}\sigma_{yy}\,\sigma_{zz} \\ \rho_{xz}\sigma_{xx}\,\sigma_{zz} & \rho_{yz}\sigma_{yy}\,\sigma_{zz} & \sigma_{zz}^2 \end{bmatrix} \tag{A3}$$

By inverting $\Sigma^{-1}$ of Eq. A2 and comparing to the standard format of $\Sigma$ as in Eq. A3, we are then able to relate how a Gaussian's size in the $x, y, z$ dimensions (governed by the three standard deviations), and shape/rotation (governed by the three correlation coefficients) are affected by the six $L_{ij}$ terms of $L$.

Unfortunately, due to the nature of our efficient decomposition, and which is optimized to primarily learn $\Sigma^{-1}$ rather than $\Sigma$, as can be seen below the six $L_{ij}$ terms are all heavily coupled and it is hard to get a physical intuition of what each term is responsible for. Nevertheless, the sensitivity graphs centered around 4 (the initial values of $L_{ij}$) do provide some insight (see Fig. A3).

It is also evident from the equations below that all six of a Gaussian's defining parameters are solely a function of $L$, and so $L$ can be thought of as the 6 Degree of Freedom matrix directly responsible for defining a 3D Gaussian's 6 DoF shape and size. Whilst the *individual* terms of $L$ are not as interpretable as the original gaussian splatting's parameterization, which used parameters $q_r, q_i, q_j, q_k, s_x, s_y, s_z$ to rotate and scale the gaussian, $L$ as a whole is still responsible for the shape and size of a Gaussian, and nothing else. We believe this slight loss in interpretability is overwhelmingly outweighed by the efficiency gains demonstrated (see Appendix A.1).

For choosing initialisation values, it is also helpful to note that generally, increasing all six $L_{ij}$ values causes $L$ and subsequently $\Sigma^{-1}$ to be larger. The covariance $\Sigma$, equal to the inverse of $\Sigma^{-1}$, will thus be smaller, and so the 3D Gaussians will start off smaller.

$$\sigma_{xx} = \frac{\sqrt{L_{12}^2 L_{23}^2 + L_{12}^2 L_{33}^4 - 2L_{12}L_{22}^2 L_{13}L_{23} + L_{22}^4 L_{13}^2 + L_{22}^4 L_{33}^4}}{L_{11}^2 L_{22}^2 L_{33}^2}$$

$$\sigma_{yy} = \frac{\sqrt{L_{23}^2 + L_{33}^4}}{L_{22}^2 L_{33}^2}$$

$$\sigma_{zz} = \frac{1}{L_{33}^2}$$

$$\rho_{xy} = \rho_{yx} = -\frac{L_{12}(L_{23}^2 + L_{33}^4) - L_{22}^2 L_{13}L_{23}}{\sqrt{L_{23}^2 + L_{33}^4}\sqrt{L_{12}^2 L_{23}^2 + L_{12}^2 L_{33}^4 - 2L_{12}L_{22}^2 L_{13}L_{23} + L_{22}^4 L_{13}^2 + L_{22}^4 L_{33}^4}}$$

$$\rho_{xz} = \rho_{zx} = -\frac{-L_{13}L_{22}^2 + L_{12}L_{23}}{\sqrt{L_{12}^2 L_{23}^2 + L_{12}^2 L_{33}^4 - 2L_{12}L_{22}^2 L_{13}L_{23} + L_{22}^4 L_{13}^2 + L_{22}^4 L_{33}^4}}$$

$$\rho_{yz} = \rho_{zy} = \frac{L_{23}}{\sqrt{L_{23}^2 + L_{33}^4}}$$

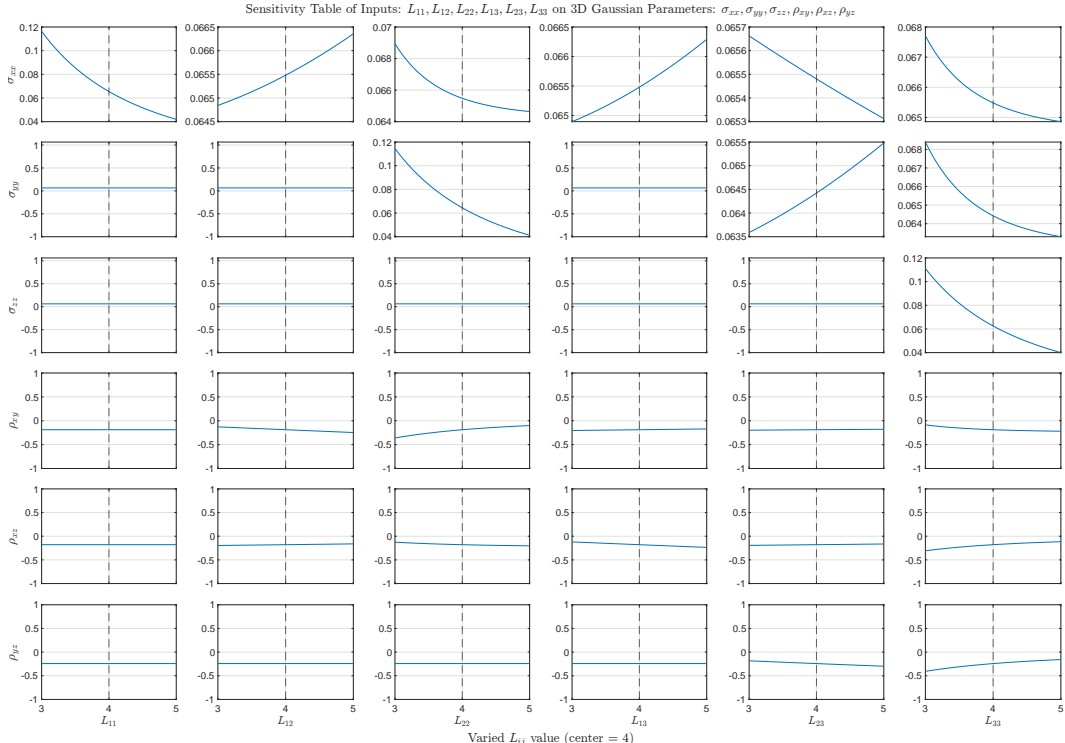

Figure A3: Sensitivity table showing the effect each $L_{ij}$ term in Eq. 11 has on a Gaussian's 6 defining parameters as it is perturbed and all other $L_{ij}$ values are held constant. Perturbations are shown here centered about 4 as that is the initialisation value of all $L_{ij}$s, however during the optimization process each $L_{ij}$ term is free to take any real value.

# B  EXPERIMENT 5.2 (3D SCANS) - ADDITIONAL RESULTS

| Input Set | Time t | Model | Training SSIM (↑) | Axial SSIM (↑) | Axial PSNR (↑) | Axial LPIPS (↓) | Coronal SSIM (↑) | Coronal PSNR (↑) | Coronal LPIPS (↓) | Sagittal SSIM (↑) | Sagittal PSNR (↑) | Sagittal LPIPS (↓) | Avg SSIM (↑) | Avg PSNR (↑) | Avg LPIPS (↓) |
|---|---|---|---|---|---|---|---|---|---|---|---|---|---|---|---|
| | | | | | | | **Test Slices, 160 Linearly Spaced:** | | | | | | | | |
| 160 Linearly spaced Axial Slices | 5 mins | ImplicitVol | 0.850±0.043 | 0.793±0.058 | 26.44±4.07 | 0.203±0.056 | 0.698±0.052 | 23.07±2.04 | 0.286±0.056 | 0.707±0.056 | 25.78±2.99 | 0.298±0.066 | 0.732±0.055 | 25.10±3.03 | 0.262±0.060 |
| | | RapidVol | 0.240±0.047 | 0.240±0.051 | 15.96±1.39 | 0.648±0.039 | 0.235±0.052 | 15.83±1.29 | 0.661±0.036 | 0.258±0.052 | 16.30±1.74 | 0.647±0.051 | 0.244±0.052 | 16.03±1.47 | 0.652±0.042 |
| | | UltraNerf | 0.531±0.038 | 0.494±0.037 | 19.39±2.35 | 0.318±0.027 | 0.412±0.049 | 16.71±1.48 | 0.412±0.028 | 0.252±0.096 | 14.47±0.54 | 0.585±0.015 | 0.386±0.061 | 16.85±1.45 | 0.438±0.024 |
| | | **UltraGauss - 100K** | 0.985±0.003 | 0.986±0.003 | **43.07±2.92** | **0.013±0.002** | **0.982±0.004** | **40.85±1.71** | **0.028±0.005** | **0.979±0.004** | **44.25±2.75** | **0.036±0.007** | 0.982±0.004 | **42.72±2.46** | **0.026±0.005** |
| | | **UltraGauss - 300K** | 0.981±0.003 | 0.982±0.003 | 40.73±1.75 | 0.031±0.003 | 0.978±0.004 | 38.96±1.20 | 0.050±0.005 | 0.978±0.004 | 42.10±1.89 | 0.055±0.006 | 0.980±0.004 | 40.60±1.62 | 0.045±0.005 |
| | | **UltraGauss - 2M** | **0.986±0.002** | **0.987±0.002** | 41.17±1.85 | 0.024±0.003 | **0.983±0.003** | 39.31±1.29 | 0.042±0.005 | **0.983±0.003** | 42.11±2.08 | 0.049±0.006 | **0.985±0.003** | 40.86±1.74 | 0.038±0.005 |
| | 20 mins | ImplicitVol | 0.901±0.032 | 0.855±0.051 | 29.57±4.19 | 0.140±0.046 | 0.704±0.047 | 23.47±1.55 | 0.265±0.052 | 0.708±0.051 | 26.22±2.90 | 0.273±0.064 | 0.756±0.050 | 26.42±2.88 | 0.226±0.054 |
| | | RapidVol | 0.807±0.080 | 0.808±0.087 | 24.67±4.88 | 0.225±0.106 | 0.789±0.087 | 23.96±3.84 | 0.261±0.096 | 0.789±0.086 | 29.61±4.13 | 0.227±0.077 | 0.795±0.086 | 26.08±4.28 | 0.238±0.093 |
| | | UltraNerf | 0.590±0.060 | 0.561±0.047 | 20.65±1.91 | 0.299±0.053 | 0.442±0.041 | 17.21±1.59 | 0.413±0.056 | 0.260±0.055 | 14.47±0.76 | 0.586±0.013 | 0.421±0.047 | 17.44±1.32 | 0.432±0.041 |
| | | **UltraGauss - 100K** | 0.990±0.002 | **0.995±0.001** | **46.61±2.88** | **0.003±0.000** | 0.987±0.003 | 43.44±1.79 | 0.016±0.003 | 0.979±0.004 | 45.51±3.19 | 0.021±0.004 | 0.986±0.003 | 44.51±2.90 | 0.016±0.003 |
| | | **UltraGauss - 300K** | **0.998±0.000** | 0.987±0.002 | 45.37±3.22 | 0.005±0.001 | **0.991±0.002** | 42.24±1.59 | **0.012±0.002** | **0.986±0.003** | 45.51±3.19 | **0.021±0.004** | **0.991±0.002** | **45.19±2.62** | **0.012±0.002** |
| | | **UltraGauss - 2M** | 0.986±0.002 | 0.986±0.002 | 44.50±1.79 | 0.024±0.003 | 0.983±0.003 | 44.50±1.79 | 0.042±0.005 | 0.983±0.003 | 44.68±4.41 | 0.023±0.004 | 0.990±0.002 | 44.50±1.79 | 0.011±0.002 |
| | Convergence | ImplicitVol | 0.960±0.015 | 0.960±0.015 | 39.36±4.38 | 0.041±0.014 | 0.687±0.046 | 23.69±1.42 | 0.243±0.041 | 0.685±0.052 | 26.79±3.29 | 0.245±0.053 | 0.778±0.038 | 29.95±3.03 | 0.176±0.036 |
| | | RapidVol | 0.982±0.004 | 0.982±0.004 | 42.01±3.72 | 0.013±0.003 | 0.975±0.005 | 40.63±2.41 | 0.022±0.003 | 0.974±0.006 | 45.82±3.45 | 0.027±0.006 | 0.977±0.005 | 42.82±3.19 | 0.021±0.004 |
| | | UltraNerf | 0.680±0.091 | 0.679±0.090 | 22.78±3.78 | 0.234±0.058 | 0.456±0.068 | 17.04±1.66 | 0.379±0.049 | 0.275±0.041 | 14.58±0.81 | 0.580±0.013 | 0.470±0.066 | 18.13±2.08 | 0.398±0.040 |
| | | **UltraGauss - 100K** | 0.991±0.002 | 0.992±0.002 | 45.75±3.22 | 0.004±0.001 | 0.988±0.003 | 42.48±1.65 | 0.015±0.003 | 0.977±0.004 | 45.70±4.09 | 0.030±0.006 | 0.986±0.003 | 44.64±2.98 | 0.016±0.003 |
| | | **UltraGauss - 300K** | 0.998±0.000 | 0.998±0.000 | 50.38±3.59 | 0.001±0.000 | 0.979±0.004 | 44.50±1.79 | 0.009±0.001 | 0.979±0.004 | 46.69±4.41 | 0.023±0.004 | 0.990±0.002 | 47.19±3.26 | 0.011±0.002 |
| | | **UltraGauss - 2M** | **0.999±0.000** | **1.000±0.000** | **54.26±2.86** | **0.000±0.000** | **0.995±0.001** | **46.29±1.68** | **0.007±0.001** | **0.990±0.002** | **49.63±3.52** | **0.011±0.002** | **0.995±0.001** | **50.06±2.69** | **0.006±0.001** |
| 80 Linearly sliced Axial Slices | 5 mins | ImplicitVol | 0.855±0.033 | 0.752±0.031 | 24.72±2.23 | 0.227±0.034 | 0.654±0.037 | 22.83±1.70 | 0.358±0.026 | 0.660±0.034 | 25.37±2.00 | 0.346±0.025 | 0.688±0.034 | 24.31±1.98 | 0.310±0.028 |
| | | RapidVol | 0.248±0.055 | 0.248±0.059 | 15.88±1.39 | 0.646±0.042 | 0.243±0.071 | 15.75±1.27 | 0.661±0.042 | 0.265±0.060 | 16.25±1.79 | 0.653±0.055 | 0.252±0.060 | 15.96±1.48 | 0.653±0.046 |
| | | UltraNerf | 0.501±0.044 | 0.428±0.085 | 18.49±0.87 | 0.337±0.032 | 0.372±0.087 | 16.25±1.86 | 0.417±0.029 | 0.232±0.071 | 14.05±1.51 | 0.564±0.031 | 0.344±0.081 | 16.26±1.41 | 0.439±0.030 |
| | | **UltraGauss - 100K** | 0.990±0.002 | 0.980±0.004 | 39.99±2.99 | 0.015±0.003 | 0.970±0.007 | 37.60±2.09 | 0.030±0.006 | 0.965±0.008 | 41.37±3.13 | 0.037±0.007 | 0.972±0.007 | 39.65±2.74 | 0.027±0.005 |
| | | **UltraGauss - 300K** | **0.992±0.001** | **0.985±0.003** | **40.99±2.34** | **0.014±0.002** | **0.977±0.006** | **38.03±2.20** | **0.025±0.004** | **0.974±0.006** | 40.70±2.85 | **0.033±0.005** | **0.978±0.005** | **39.90±2.47** | **0.024±0.004** |
| | | **UltraGauss - 2M** | 0.986±0.002 | 0.981±0.003 | 39.63±1.91 | 0.032±0.004 | 0.974±0.005 | 37.36±1.82 | 0.050±0.005 | **0.975±0.005** | **40.28±2.41** | 0.056±0.006 | 0.977±0.004 | 39.09±2.05 | 0.046±0.005 |
| | 20 mins | ImplicitVol | 0.929±0.013 | 0.769±0.039 | 26.09±2.63 | 0.172±0.020 | 0.659±0.052 | 23.34±1.75 | 0.341±0.031 | 0.659±0.050 | 26.48±2.64 | 0.331±0.037 | 0.695±0.047 | 25.30±2.34 | 0.281±0.029 |
| | | RapidVol | 0.762±0.153 | 0.745±0.168 | 23.88±5.07 | 0.261±0.119 | 0.723±0.169 | 23.06±3.90 | 0.306±0.111 | 0.721±0.171 | 27.17±6.56 | 0.289±0.127 | 0.730±0.169 | 24.71±5.18 | 0.285±0.119 |
| | | UltraNerf | 0.611±0.091 | 0.578±0.097 | 21.70±2.35 | 0.307±0.048 | 0.445±0.059 | 16.96±1.12 | 0.406±0.046 | 0.256±0.049 | 14.23±0.65 | 0.584±0.028 | 0.426±0.068 | 17.63±1.37 | 0.432±0.041 |
| | | **UltraGauss - 100K** | 0.994±0.001 | **0.985±0.003** | **41.45±3.09** | **0.009±0.002** | 0.961±0.009 | 38.62±2.10 | 0.023±0.005 | 0.949±0.010 | 40.99±4.00 | 0.061±0.010 | 0.975±0.006 | 38.60±2.99 | 0.043±0.007 |
| | | **UltraGauss - 300K** | **0.998±0.000** | 0.981±0.003 | 39.63±1.91 | 0.032±0.004 | **0.975±0.006** | **38.62±2.10** | **0.023±0.005** | **0.965±0.008** | **40.73±2.83** | **0.032±0.006** | **0.977±0.004** | **40.73±2.83** | **0.021±0.004** |
| | | **UltraGauss - 2M** | 0.986±0.002 | 0.981±0.003 | 39.63±1.91 | 0.032±0.004 | 0.974±0.005 | 37.36±1.82 | 0.050±0.005 | 0.975±0.005 | 40.28±2.41 | 0.056±0.006 | 0.977±0.005 | 39.09±2.05 | 0.046±0.005 |
| | Convergence | ImplicitVol | 0.969±0.008 | 0.745±0.042 | 26.13±2.60 | 0.171±0.020 | 0.632±0.054 | 22.97±1.84 | 0.353±0.030 | 0.632±0.052 | 26.48±2.77 | 0.353±0.039 | 0.670±0.049 | 25.15±2.38 | 0.292±0.029 |
| | | RapidVol | 0.956±0.012 | 0.924±0.014 | 32.13±3.23 | 0.073±0.019 | 0.898±0.018 | 30.60±2.53 | 0.137±0.023 | 0.895±0.020 | 35.81±3.05 | 0.134±0.020 | 0.906±0.017 | 32.85±2.94 | 0.115±0.021 |
| | | UltraNerf | 0.694±0.107 | 0.688±0.102 | 23.42±3.83 | 0.217±0.055 | 0.464±0.062 | 17.07±1.19 | 0.440±0.026 | 0.277±0.039 | 14.64±0.77 | 0.574±0.013 | 0.476±0.068 | 18.37±1.93 | 0.384±0.033 |
| | | **UltraGauss - 100K** | 0.994±0.001 | 0.974±0.001 | 38.59±3.16 | 0.019±0.003 | 0.961±0.009 | 36.22±1.81 | 0.051±0.010 | 0.949±0.010 | 40.99±4.00 | 0.061±0.010 | 0.961±0.008 | 38.60±2.99 | 0.043±0.007 |
| | | **UltraGauss - 300K** | 0.999±0.000 | 0.982±0.004 | 40.86±3.64 | 0.012±0.002 | 0.957±0.009 | 37.71±2.16 | 0.032±0.006 | 0.957±0.009 | 42.23±3.74 | 0.042±0.007 | 0.970±0.007 | 40.27±3.18 | 0.029±0.005 |
| | | **UltraGauss - 2M** | **1.000±0.000** | **0.991±0.002** | **44.40±4.36** | **0.005±0.001** | **0.984±0.004** | **40.24±2.17** | **0.014±0.003** | **0.980±0.004** | **45.89±3.91** | **0.018±0.003** | **0.985±0.003** | **43.51±3.15** | **0.012±0.002** |
| 80 Linearly spaced Axial Slices with ±5° Jitter | 5 mins | ImplicitVol | 0.893±0.018 | 0.801±0.038 | 26.98±2.81 | 0.203±0.026 | 0.786±0.042 | 26.48±1.84 | 0.217±0.028 | 0.794±0.042 | 29.92±2.50 | 0.237±0.033 | 0.794±0.041 | 27.79±2.38 | 0.219±0.029 |
| | | RapidVol | 0.265±0.056 | 0.253±0.060 | 16.00±1.42 | 0.640±0.042 | 0.246±0.061 | 15.87±1.30 | 0.659±0.041 | 0.268±0.062 | 16.36±1.81 | 0.645±0.052 | 0.256±0.061 | 16.08±1.51 | 0.648±0.045 |
| | | UltraNerf | 0.508±0.032 | 0.466±0.067 | 19.02±0.95 | 0.374±0.034 | 0.414±0.043 | 17.08±1.62 | 0.440±0.026 | 0.269±0.028 | 14.69±0.17 | 0.600±0.015 | 0.383±0.046 | 16.93±0.92 | 0.471±0.025 |
| | | **UltraGauss - 100K** | 0.994±0.000 | 0.939±0.007 | 33.34±1.28 | 0.070±0.008 | 0.931±0.009 | 30.98±1.01 | 0.065±0.007 | 0.934±0.009 | 31.60±1.05 | 0.064±0.006 | 0.935±0.008 | 31.97±1.11 | 0.066±0.007 |
| | | **UltraGauss - 300K** | **0.994±0.001** | **0.969±0.007** | **37.23±2.75** | **0.044±0.005** | **0.964±0.008** | **36.92±1.67** | **0.043±0.004** | **0.969±0.007** | **39.20±2.44** | **0.039±0.004** | **0.967±0.007** | **37.78±2.29** | **0.042±0.004** |
| | | **UltraGauss - 2M** | 0.990±0.001 | 0.965±0.006 | 36.39±2.52 | 0.067±0.008 | 0.961±0.008 | 36.08±1.52 | 0.074±0.006 | 0.967±0.007 | 38.55±2.30 | 0.067±0.006 | 0.964±0.007 | 37.00±2.12 | 0.069±0.006 |
| | 20 mins | ImplicitVol | 0.950±0.011 | 0.855±0.028 | 29.47±2.95 | 0.134±0.019 | 0.835±0.032 | 29.02±1.81 | 0.155±0.025 | 0.838±0.031 | 32.35±3.14 | 0.171±0.031 | 0.843±0.031 | 30.28±2.63 | 0.154±0.025 |
| | | RapidVol | 0.853±0.051 | 0.817±0.056 | 24.94±4.70 | 0.225±0.091 | 0.796±0.056 | 24.37±3.58 | 0.266±0.082 | 0.795±0.056 | 29.70±3.80 | 0.229±0.063 | 0.803±0.056 | 26.34±4.03 | 0.240±0.079 |
| | | UltraNerf | 0.625±0.051 | 0.477±0.040 | 18.97±0.73 | 0.323±0.035 | 0.401±0.046 | 16.88±1.39 | 0.397±0.022 | 0.235±0.041 | 13.74±1.59 | 0.576±0.027 | 0.371±0.042 | 16.53±1.07 | 0.432±0.028 |
| | | **UltraGauss - 100K** | 0.997±0.001 | 0.929±0.007 | 31.82±0.94 | 0.078±0.006 | 0.905±0.021 | 29.55±0.48 | 0.088±0.008 | 0.918±0.010 | 29.98±0.54 | 0.092±0.009 | 0.921±0.009 | 30.45±0.65 | 0.086±0.008 |
| | | **UltraGauss - 300K** | **0.999±0.000** | 0.936±0.005 | 32.50±1.21 | 0.066±0.007 | 0.917±0.010 | 29.13±0.95 | 0.059±0.007 | 0.929±0.008 | 29.55±0.93 | 0.065±0.006 | 0.933±0.006 | 30.31±0.92 | **0.061±0.006** |
| | | **UltraGauss - 2M** | 0.990±0.001 | **0.965±0.006** | **36.39±2.52** | **0.067±0.008** | **0.961±0.008** | **36.08±1.52** | **0.074±0.006** | **0.967±0.007** | **38.55±2.30** | **0.067±0.006** | **0.964±0.007** | **37.00±2.12** | 0.069±0.006 |
| | Convergence | ImplicitVol | 0.977±0.007 | 0.864±0.028 | 30.42±2.99 | 0.102±0.019 | 0.842±0.033 | 30.07±1.87 | 0.143±0.025 | 0.841±0.033 | 33.72±3.19 | 0.161±0.032 | 0.849±0.031 | 31.41±2.68 | 0.135±0.025 |
| | | RapidVol | 0.973±0.006 | 0.925±0.016 | 32.28±3.70 | 0.074±0.014 | 0.903±0.022 | 31.92±2.53 | 0.127±0.019 | 0.903±0.022 | 36.09±3.49 | 0.119±0.022 | 0.911±0.019 | 33.43±3.24 | 0.107±0.018 |
| | | UltraNerf | 0.710±0.038 | 0.610±0.078 | 20.91±2.17 | 0.265±0.036 | 0.458±0.038 | 17.30±1.04 | 0.373±0.021 | 0.264±0.043 | 14.63±0.85 | 0.584±0.010 | 0.444±0.053 | 17.61±1.35 | 0.407±0.022 |
| | | **UltraGauss - 100K** | 0.997±0.001 | 0.929±0.007 | 29.98±0.54 | 0.092±0.009 | 0.918±0.010 | 29.55±0.48 | 0.088±0.008 | 0.918±0.010 | 29.98±0.54 | 0.092±0.009 | 0.921±0.009 | 30.45±0.65 | 0.086±0.008 |
| | | **UltraGauss - 300K** | 0.999±0.000 | 0.934±0.006 | 32.34±1.26 | 0.065±0.007 | 0.927±0.008 | 29.13±0.95 | 0.059±0.007 | 0.929±0.008 | 29.55±0.93 | 0.065±0.006 | 0.930±0.007 | 30.34±1.05 | 0.063±0.007 |
| | | **UltraGauss - 2M** | **1.000±0.000** | **0.942±0.004** | **33.21±1.22** | **0.064±0.007** | **0.937±0.004** | 29.12±0.78 | **0.054±0.006** | **0.941±0.004** | 29.79±0.91 | **0.056±0.005** | **0.940±0.004** | **30.71±0.97** | **0.058±0.006** |

Figure B1: Reconstruction results for 6 different models on 3 different training sets, shown at 3 time points. ↑ indicates higher is better. "UltraGauss - N" indicates UltraGauss with N starting Gaussians. Best scores are highlighted in bold.

| Reconstruction Time (mins) | Rendered Image | Absolute Difference Image |
|---|---|---|
| Ground Truth |  | N/A |
| $t = 0.5$ |  |  |
| $t = 1$ |  |  |
| $t = 2$ |  |  |
| $t = 3$ |  |  |
| $t = 4$ |  |  |
| $t = 5$ |  |  |
| $t = 10$ |  |  |
| $t = 15$ |  |  |
| $t = 20$ |  |  |

Figure B2: Progression of UltraGauss as a 3D reconstruction is formed from 80 Axial Slices (sampled from a 20 GW 3D fetal scan). The image requested to be rendered image is an orthogonal, mid-coronal cross-section. Clinicians were then asked to chose whether each one of these was a real or "AI Generated" scan. The results to this are in Fig. 5

## C  EXPERIMENT 5.3 (CLINICIANS' SURVEY)

### C.1  THE SURVEY

Below is the questionnaire which Clinicians answered:

PRELIMINARY
1. What is your current position?
2. How many years of medical experience do you have?
3. On a scale of 1 to 5, how would you rate your confidence in assessing **Fetal Brains in Ultrasound**? (5 is very confident, 1 is not at all)

PART I
Participants answered a random selection of 10 pair-wise comparisons. An example of one is shown below:
*Please **choose the most realistic** ultrasound image out of the two.*

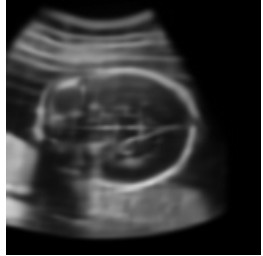
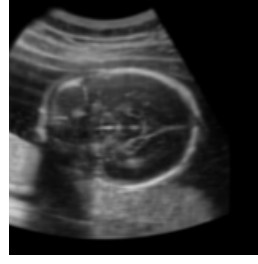
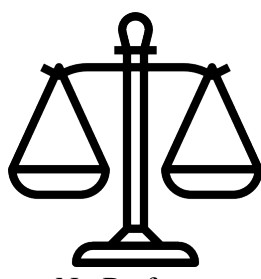

    ○ Option 1       ○ Option 2      ○ No Preference

PART II
Participants then answered 10 of the following questions. The order of these 10 questions and the order of the two options within each question was randomly shuffled for each participant.
*The ultrasound scans below show mid-sagittal views of the same fetal brain. Some are real scans acquired using standard ultrasound equipment, while others are AI-generated reconstructions. For each image, please **identify whether the scan is real or AI-generated**.*

*Real ultrasound scan, or AI reconstruction?*

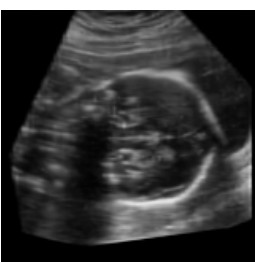

○ Real
○ AI-generated

## C.2 CLINICIANS' SURVEY ADDITIONAL RESULTS

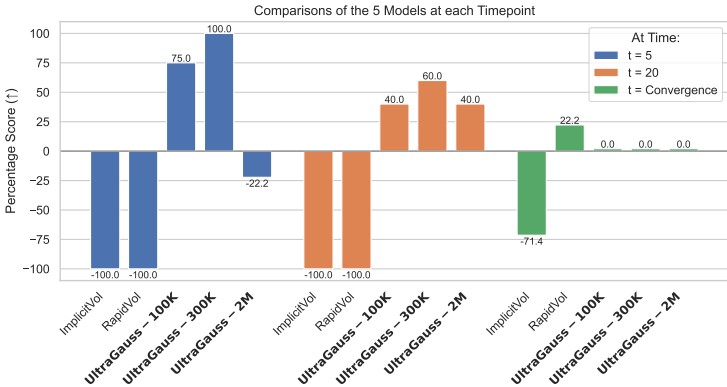

Figure C2: Survey results: Comparison of all 5 models at each time point. Actively choosing a model gave it 1 point, not choosing it -1, and choosing "no preference" gave it 0 points. Each model's score was then divided by the total number of points it could have gained, to give a percentage. Thus +100% means that all participants actively selected and preferred that model. -100% means that all participants actively disliked this model (and selected the other one). 0% means that the model is neither liked or disliked, as the participant chose the "no preference" option.

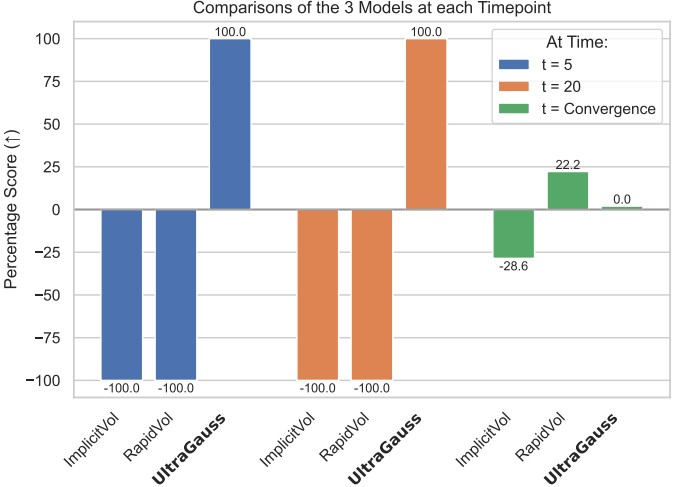

Figure C3: Survey results: Comparison of ImplicitVol, RapidVol, and the better of the three Ultra-Gauss Models at each time point. The same scoring system applies as Fig. C2

# D EXPERIMENT 5.4 (US VIDEOS)

## D.1 FURTHER EXPERIMENTAL SETUP DETAILS

In this experiment we use ultrasound cinesweep videos acquired using standard sensorless probes. As such, the poses of their frames have to first be estimated using a suitable pose estimation that works with sensorless probes (e.g. (Ramesh et al., 2024; Di Vece et al., 2024; Yeung et al., 2022)). 20% of the frames are then randomly removed and witheld for testing. The remaining frames (blue frames in Input Video on Fig. D1) and their corresponding poses (blue probes/planes) are then inputted into UltraGauss to form a 3D reconstruction of the individual fetal brain within 5 minutes. Images are then sampled from the reconstructed volume at the poses of the withheld video frames (green probes/planes), and are shown in green in the Output Video tape.

A similar method also applies to forming 3D reconstructions from Dataset A. Rather than the blue Input Views being video frames and their predicted poses, they are instead slices sampled from the volumetric scans of Dataset A and their corresponding ground truth poses.

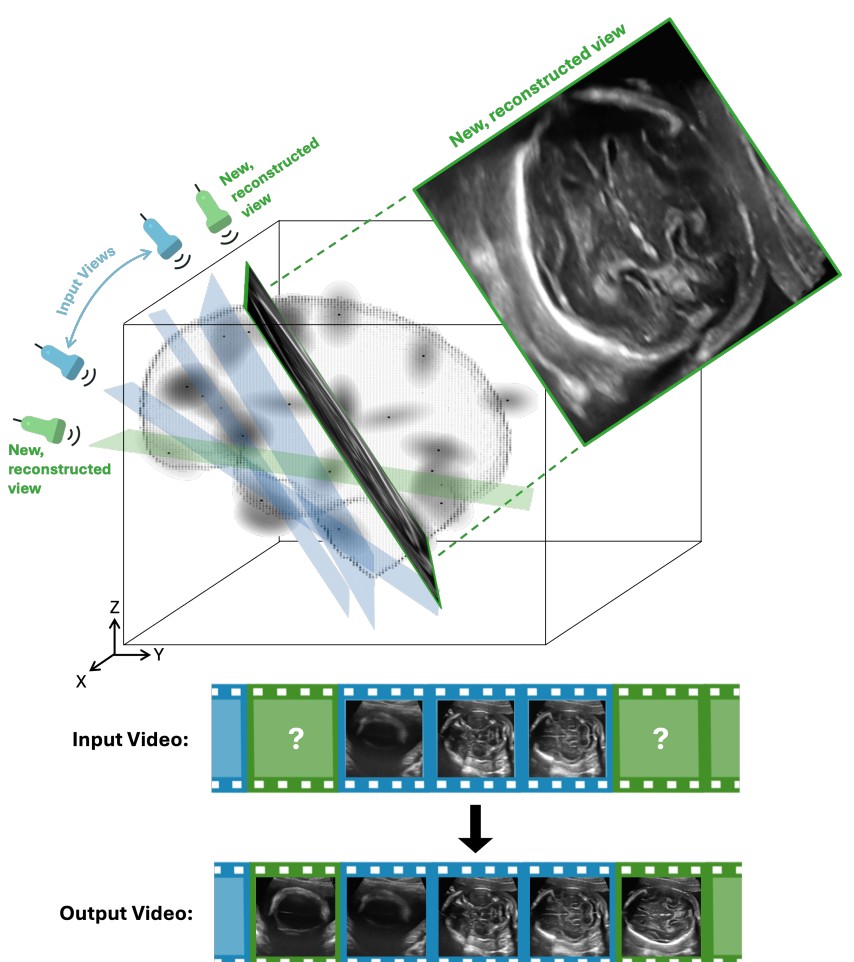

Figure D1: Input images of the fetal brain are acquired (blue Input Video frames) and their poses estimated (blue probes/planes). UltraGauss then uses these to form a total 3D reconstruction within 5 minutes. A few of the 3D Gaussians are shown as gray ellipses. Cross-sectional views at previously unseen poses can then be sampled from the reconstructed 3D volume to form a complete cinesweep or be viewed individually. These can be seen in the green Output Video cinefilm frames.

## D.2 ADDITIONAL RESULTS FROM VIDEOS

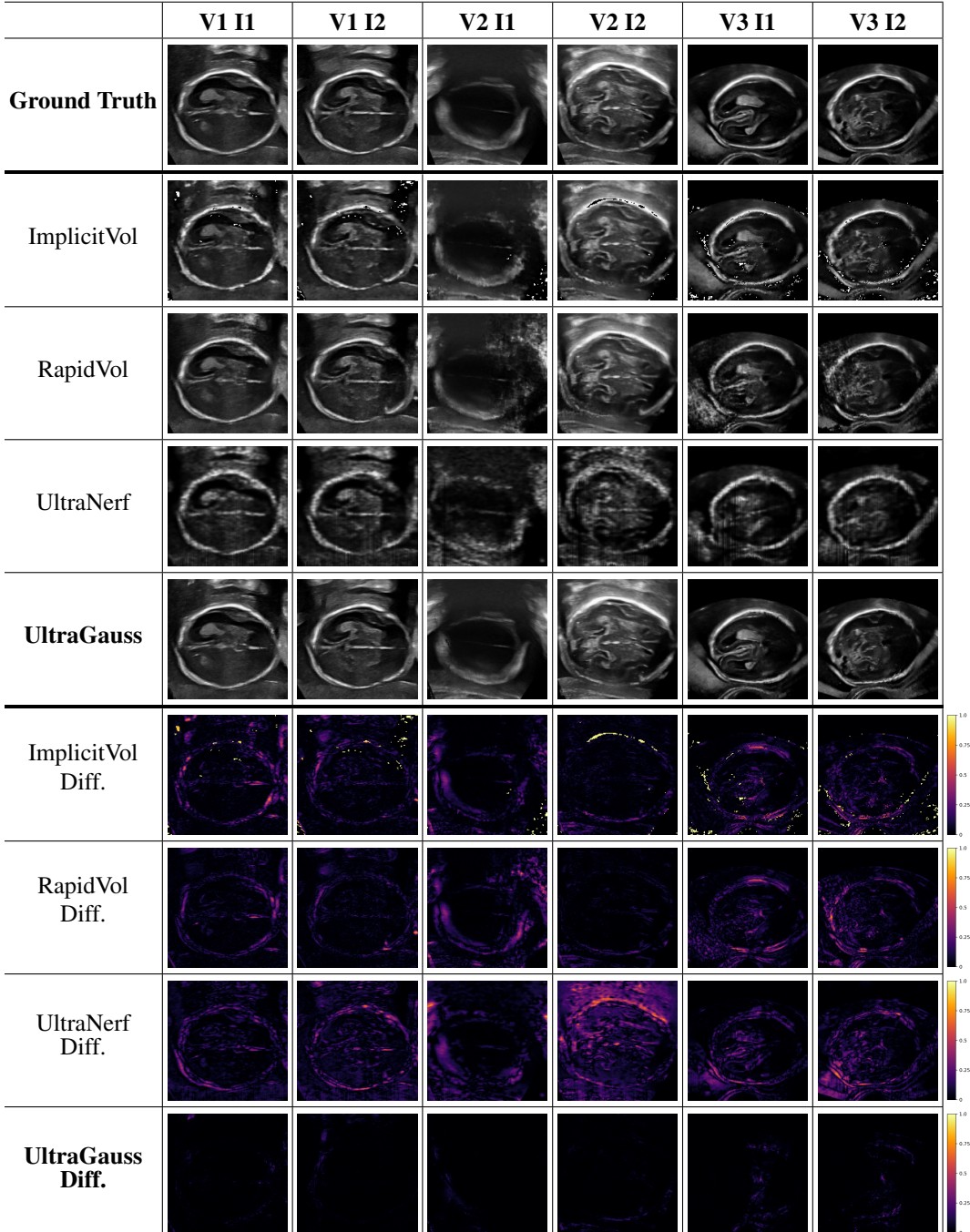

Continued below...

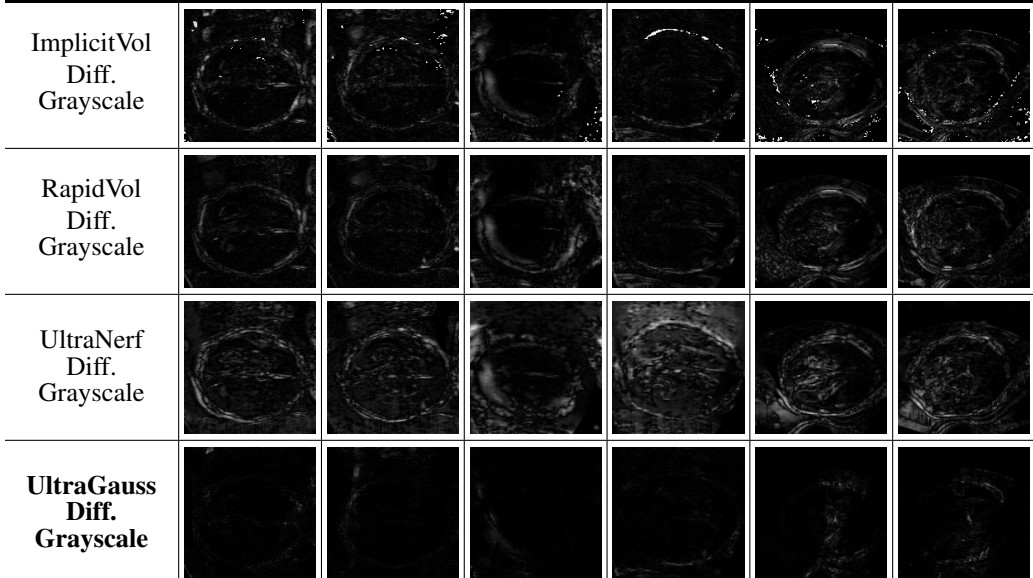

Figure D2: For each video (V$n$), two ground truth images (I$j$) from the test set are shown, alongside the corresponding image generated by ImplicitVol, RapidVol, UltraNerf and UltraGauss - 300K (ours). The absolute difference (diff.) between the predicted and ground truth image is also shown, in both grayscale and the perceptually uniform inferno colored heatmap. The inferno color bar ranging from 0.0 to 1.0 is shown on the right-hand side. A pure black diff. image means perfect similarity, and so the darker the diff image the better.

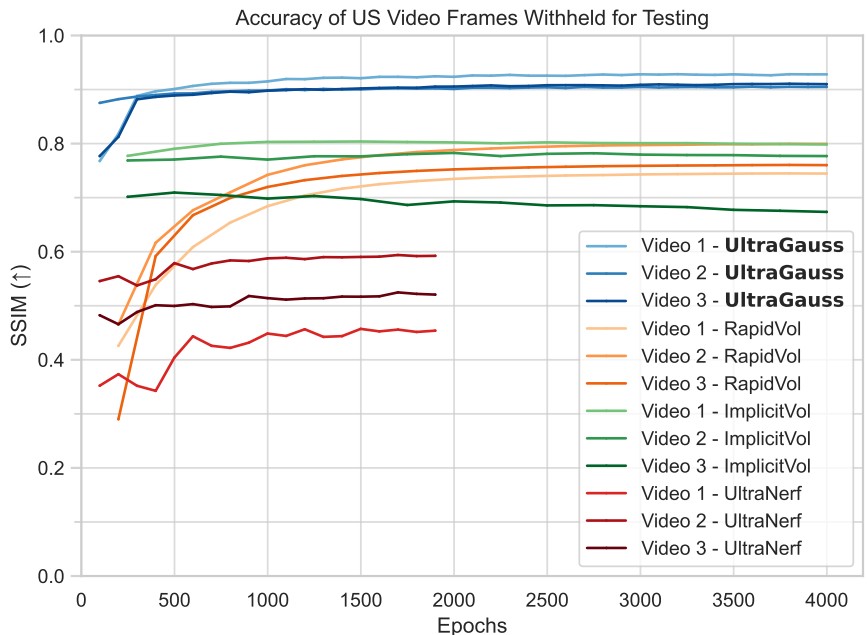

Figure D3: Test accuracy over the course of reconstruction/training for UltraGauss - 300K (ours, shown in blue), UltraNerf, RapidVol and ImplicitVol, reported on 3 ultrasound cinesweeps. Higher SSIM is better.

# E ANATOMICAL ACCURACY OF ULTRAGAUSS' 3D RECONSTRUCTIONS

In Figs. E1 and E2 we overlay segmentation masks (produced automatically using a convolutional neural network (Hesse et al., 2022; Namburete et al., 2023)) over the UltraGauss reconstructions in the hardest of our 3 scenarios - using 80 Axial slices with $\pm5°$ Jitter (to mimic the sonographer's natural hand motion). We also segment the original 3D volume as a benchmark. We show mid-axial, mid-coronal and mid-sagittal views, at both 20 and 26 Gestational Weeks (no segmentation masks were available for 14 GW). All the UltraGauss reconstructions show high anatomical accuracy.

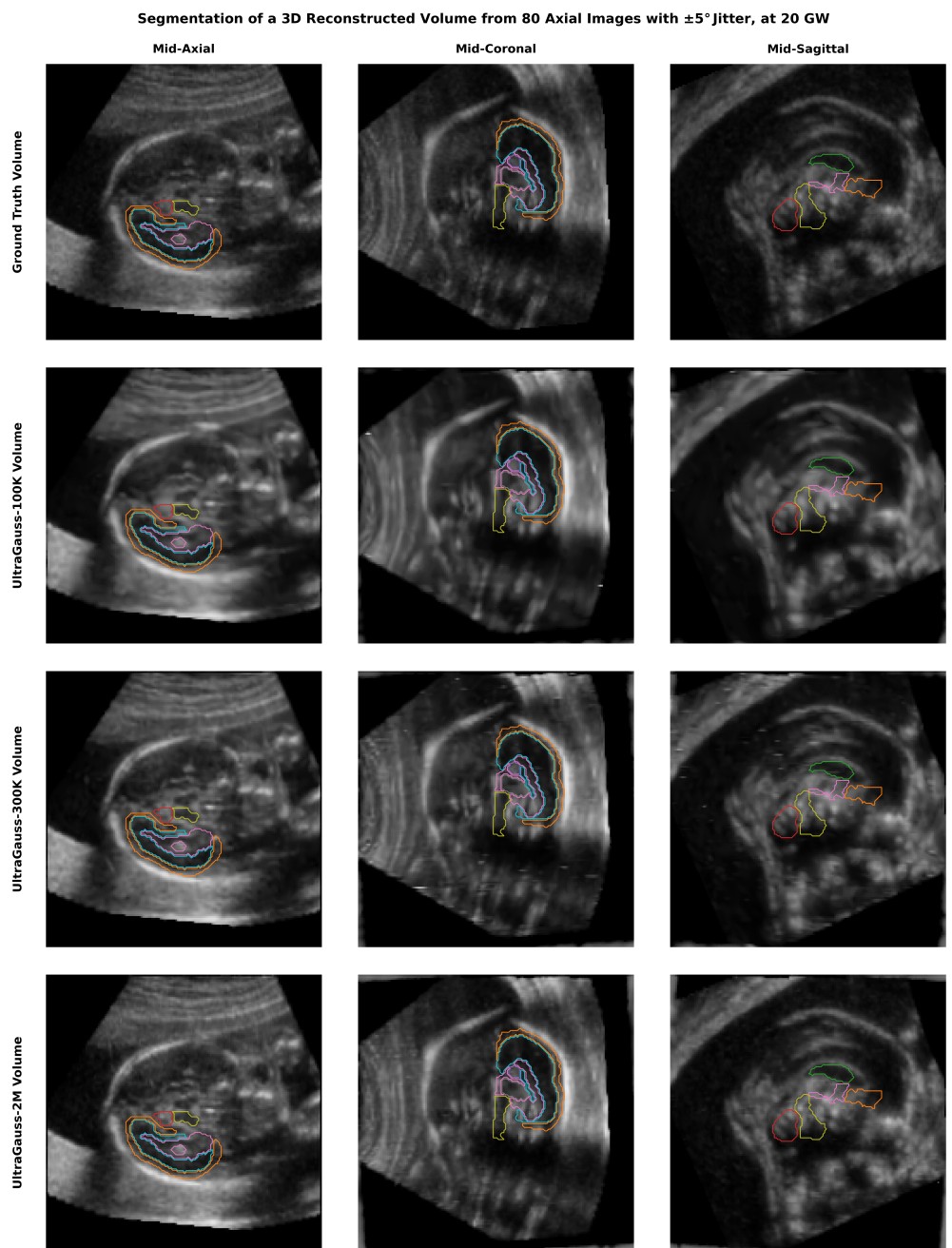

Figure E1: Segmentation mask applied to the Original and UltraGauss-reconstructed volumes of a 20 GW fetus. The input to the reconstructions was 80 Axial slices with $\pm5°$ of rotational jitter in the plane.

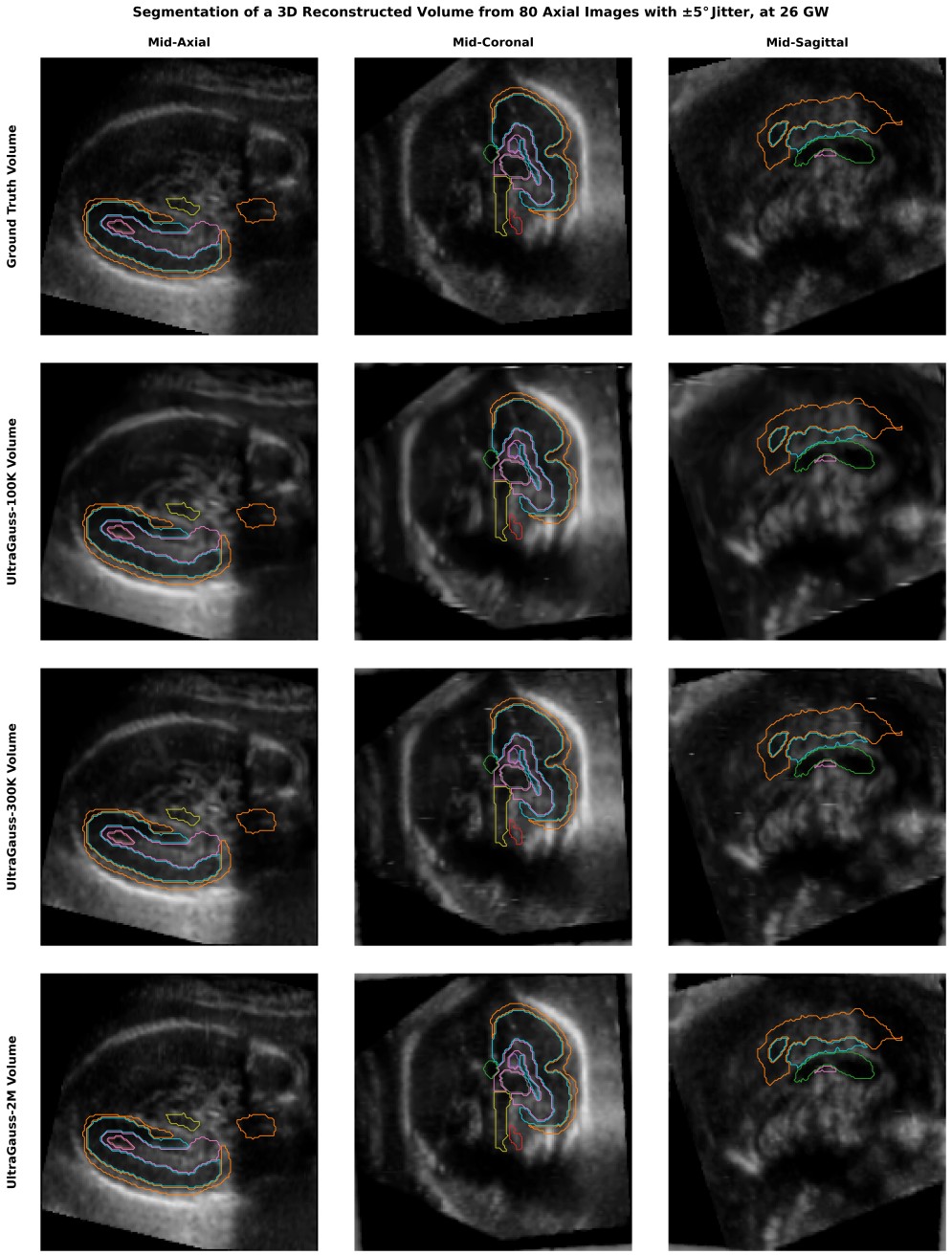

Figure E2: Segmentation mask applied to the Original and UltraGauss-reconstructed volumes of a 26 GW fetus. The input to the reconstructions was 80 Axial slices with $\pm 5°$ of rotational jitter in the plane.

## F  PRELIMINARY RESULTS ON ULTRASOUND CINESWEEPS OF FOREARMS

To further demonstrate that UltraGauss is not organ-specific and can be applied to general 2D ultrasound cinesweeps, we also validate on the TUS-REC Dataset (Li et al., 2025). This dataset of adult forearm ultrasound cinesweeps was acquired using an Ultrasonix machine equipped with a curvlinear probe (4DC7-3/40). A NDI Polaris Vicra optical tracking system was used to provide ground truth poses, and standard ultrasound post-processing speckle reduction was **not** applied. S shape trajectory scans and poses were used as input into UltraGauss, and then novel views were sampled from the 3D reconstruction at the poses of the straight line trajectory scans. Preliminary results can be found in Fig. F1. It is evident that UltraGauss still forms good 3D reconstructions, and that the main structural content and anatomy is modelled accurately. Most of the pixel-wise difference seen in Fig. F1 is in areas with a lot of fine speckle and which would require many, extremely small Gaussians. In practice there would not be this much speckle in the ultrasound cinesweeps, as post-processing speckle reduction would be applied, and so the reconstructions would be even better. It is only in this dataset the authors intentionally chose not to do this, in order to help the participants in their pose-prediction challenge.

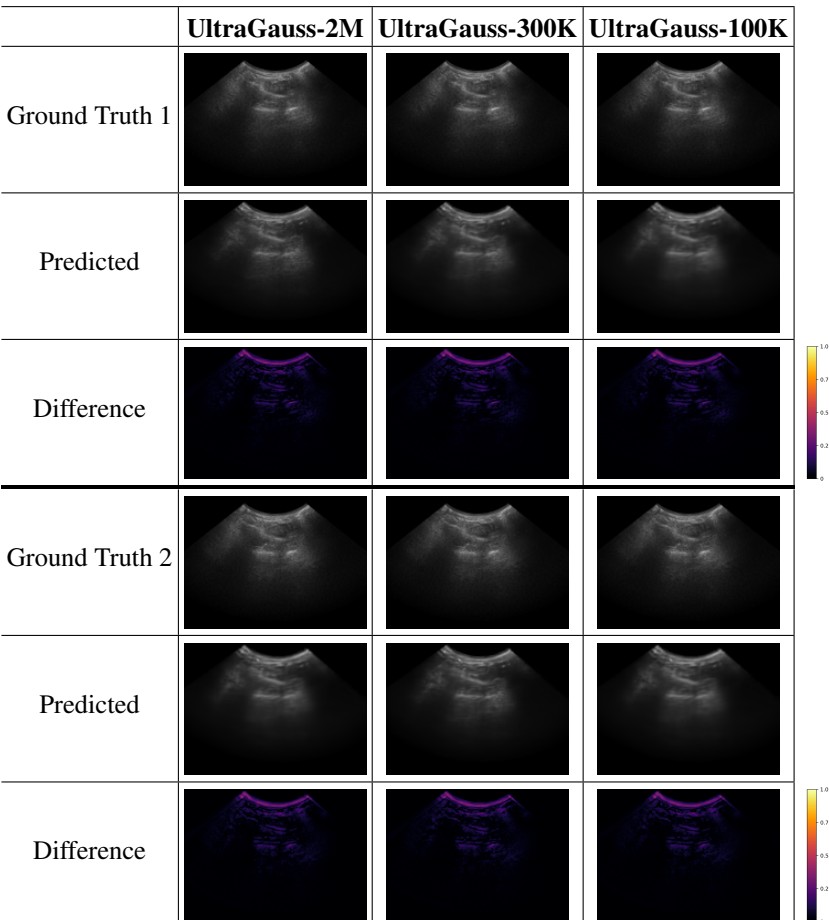

Figure F1: UltraGauss - {2M, 300K, 100K} was applied to form a 3D reconstruction from an input cinesweep of a human participant's forearm. Two novel views were then sampled and are shown here alongside their ground truth. The pixel-wise absolute difference between the predicted and ground truth images are shown in the perceptually uniform `inferno` colored heatmap, with the color bar ranging from 0.0 to 1.0 shown on the right-hand side. A pure black difference image means perfect similarity, and so the darker the difference image the better.

# G  ABLATIONS

## G.1  $c$ AND $\alpha$ OR ONE COMBINED PARAMETER?

Since ultrasound is a grayscale modality, requiring only one color channel, it is possible to combine each Gaussian's single-channel color $c_i$ (which ranges from 0 to 1) and opacity $\alpha_i$ (also ranges from 0 to 1) into a single learnable parameter, say $\rho_i$, as for example is done by Zha et al. (2024). We could then multiply this by Gaussian $i$'s probability at point $\mathbf{x}$. This would then be repeated for all the other Gaussians and they can then either be summed together (but stopping if the sum reaches 1), or have the average taken. Since we do not splat Gaussians onto an image plane and sort them by depth from front to back, it is not obvious how to decide which 3D Gaussians to exclude from the sum if it reaches 1, so we would have to take the average. Expressed formally this potential approach is:

$$\hat{\rho}_i\left(x\right) = \rho_i \exp\left(-\frac{1}{2}\underbrace{\left(x_{|0} - \mu_i^{\text{3D}}\right)^T \left(\Sigma_i^{\text{3D}}\right)^{-1}\left(x_{|0} - \mu_i^{\text{3D}}\right)}_{\text{3D squared Mahalanobis distance}}\right)$$

$$\hat{c}(x) = \frac{\sum_i^n \hat{\rho}_i(x)}{n} \tag{G1}$$

Instead, we propose a formulation as in Eq. 7. Ignoring $\alpha_{BG}$ and $c_{BG}$ for now, we are essentially proposing:

$$\hat{c}\left(x\right) = \frac{\sum_i^n \hat{\alpha}_i\left(x\right)c_i}{\sum_i^n \hat{\alpha}_i\left(x\right)} \tag{G2}$$

which mathematically is a *weighted* average. Compare this with the *arithmetic* average when one parameter is used. The former is naturally more powerful, and Table G1 confirms this. In our application, intuitively, $\alpha_i$ dictates the importance of each Gaussian, and how strongly its color should feature in the final pixel color $\hat{c}$.

Table G1: Using $c_i$ and $\alpha_i$ vs. only one parameter. Scores are the avg. across all three standard test views, for four 20 week fetuses from Dataset A.

| | Input Set = 80 Axial Slices | | | Input Set = 160 Axial Slices | | |
|---|---|---|---|---|---|---|
| | SSIM ($\uparrow$) | PSNR ($\uparrow$) | LPIPS ($\downarrow$) | SSIM ($\uparrow$) | PSNR ($\uparrow$) | LPIPS ($\downarrow$) |
| One Param. | $0.931 \pm 0.012$ | $3.42 \pm 0.100$ | $0.097 \pm 0.026$ | $0.947 \pm 0.011$ | $3.66 \pm 0.120$ | $0.085 \pm 0.026$ |
| $c_i$ and $\alpha_i$ | $\mathbf{0.967 \pm 0.012}$ | $\mathbf{3.70 \pm 0.193}$ | $\mathbf{0.032 \pm 0.015}$ | $\mathbf{0.990 \pm 0.009}$ | $\mathbf{4.42 \pm 0.382}$ | $\mathbf{0.011 \pm 0.009}$ |

## G.2 THE IMPACT OF SHADOW MODELLING AND DENSIFICATION

Fig. G2 visibly shows the beneficial impact adaptive Gaussian densification and sparsification, once modified and implemented appropriately, can have. Greater sharpness and speckle can be seen in the top quarter of the images (a) when Densification is turned on (c.f. 2nd column with 3rd column). When Shadow Modelling is turned on, view-dependent acoustic shadows become more realistic. Comparing columns 3 and 4, one can see that shadows or dark patches no longer appear where they shouldn't (b), shadows no longer bulge out and curve, but rather become straight (c & e), and they also no longer abruptly stop (d).

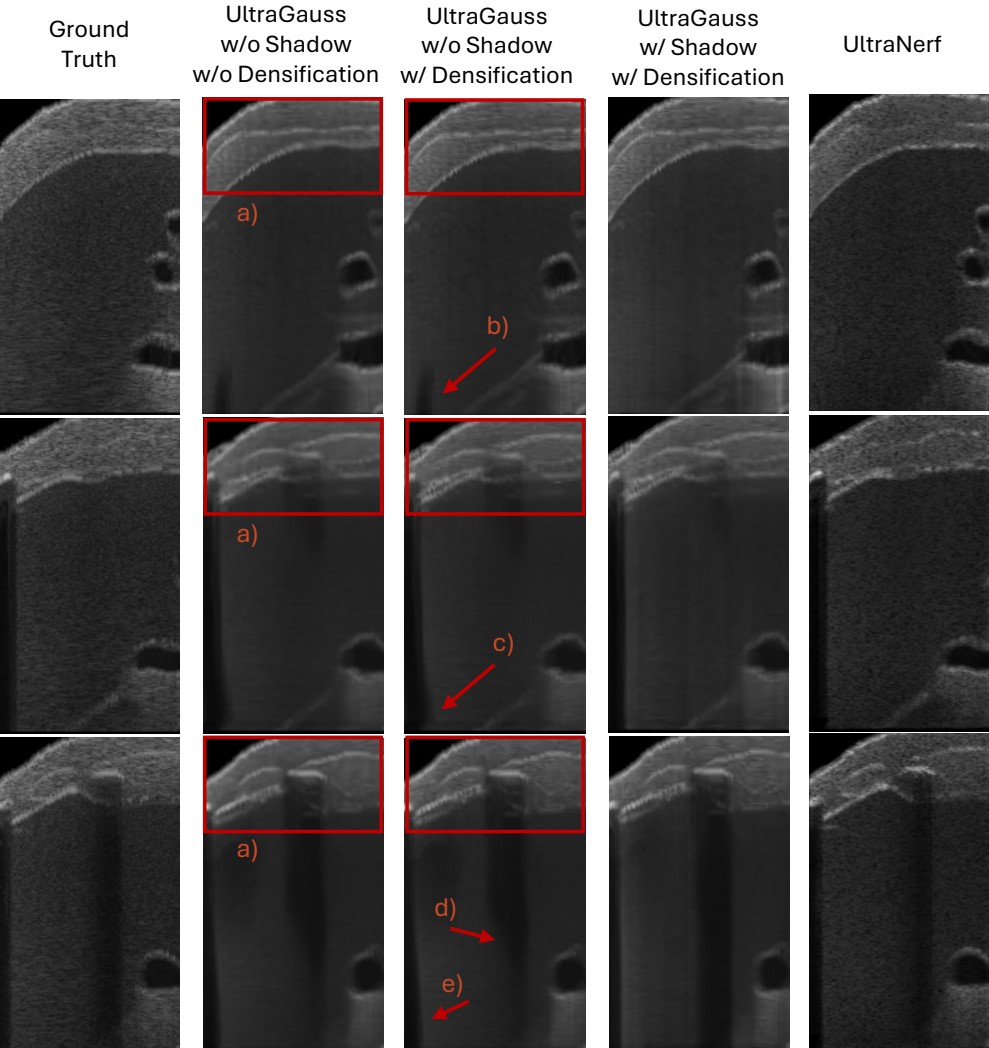

Figure G2: Test-set images rendered using UltraGauss-300K, with (w/) and without (w/o) shadow modelling or adaptive Gaussian densification & sparsification. A fully ultrasound physics based model, UltraNerf (Wysocki et al., 2023) serves as a baseline.

### G.3 THE EFFECT OF A GAUSSIAN'S PROBABILITY DENSITY $p$ VALUE

We expected that as the gaussian probability density threshold value was increased, the size of a Gaussian's ellipsoid of influence would increase, and so more pixels would consider it to be "close" and accumulate it. Applied to all $N$ Gaussians, this would mean more atomic additions and so increase the time taken, but lead to potentially higher accuracy as each pixel's final grayscale value is now controlled by more Gaussians. To confirm this we conducted an ablation on UltraGauss-300K applied to one of the 3D scans from Dataset A, and varied the $p\%$ value (see Eq. 13). Metrics are averaged across all 3 test views. Our ablation confirmed our expectations and as such we chose $p = 95\%$ as a good compromise between the number of Gaussians accumulated each iteration and reconstruction accuracy (considering all three metrics).

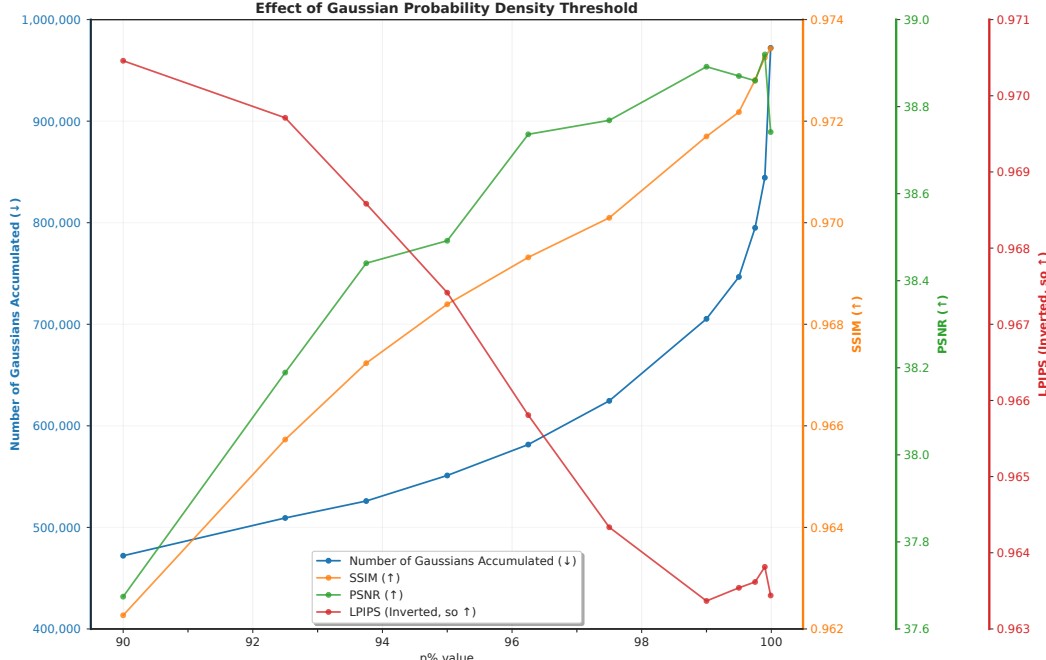

Figure G3: Ablation study on UltraGauss in which the $p$ value of the Gaussians' bounding probability density ellipsoids are increased. Metrics are averaged across all 3 test views.

## G.4 THE EFFECT OF POSE ESTIMATION ERROR

To attempt to disentangle the error as a consequence of having inaccurate poses from the error due to the 2D-to-3D reconstruction process, we conducted the following ablation. For 80 linearly spaced axial slices sampled from the 3D volumes of Dataset A, rather than providing these images along with their ground truth poses as we did in Sec. 5.2, we now intentionally add some uniform random noise to the poses, to simulate UltraGauss being given inaccurate poses as would be the case when using the cinesweeps in our end-to-end pipeline (Sec. 5.4).

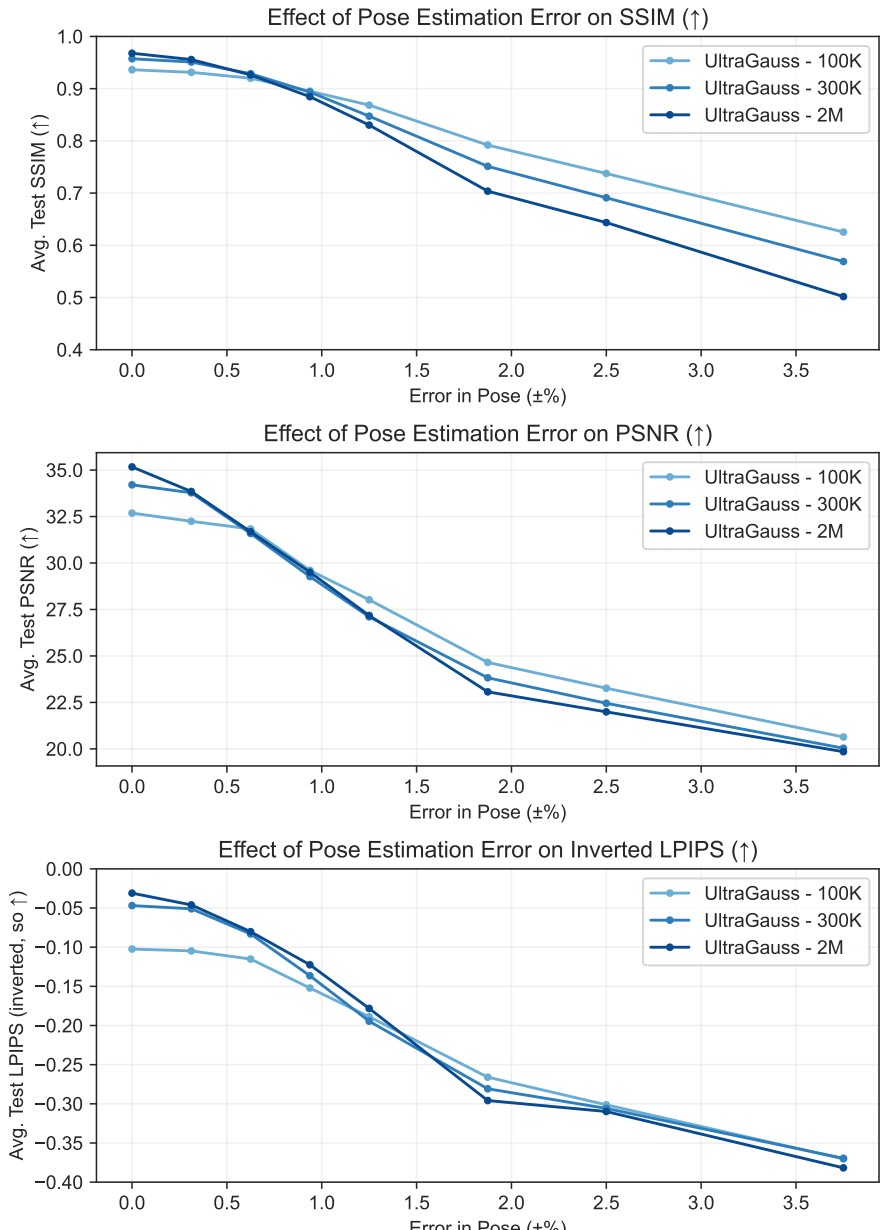

Figure G4: Ablation study on how the different variants of UltraGauss cope as error in the pose of the reconstruction input images increases. LPIPS is shown inverted here, so that like SSIM and PSNR, a higher value signifies better accuracy.

From Fig. G4 it can be seen as is typical with reconstruction methods which are overfitted to a particular scene and are not pre-trained, they do require fairly accurate registration of images to a fixed co-ordinate system. For UltraGauss the error limit is about $1.5\%$ before reconstruction accuracy degrades.

## G.5 THE EFFECT OF SCAN COVERAGE

Here we investigate how well UltraGauss copes with incomplete scan coverage. We train on less and less axial slices, and report the average SSIM, PSNR and Inverted LPIPS averaged across the novel axial, coronal and sagittal views generated. Generally, quality is good until around 25% coverage of the subject, beyond which it drops significantly. Since UltraGauss is not pre-trained, and like most other 3D reconstruction methods over-optimizes to the specific scene, this behaviour is somewhat expected. Crucially, it also ensures that UltraGauss does not hallucinate or create non-existent anatomy, something that is crucial for use in medical settings and we actively considered when tuning our heuristics.

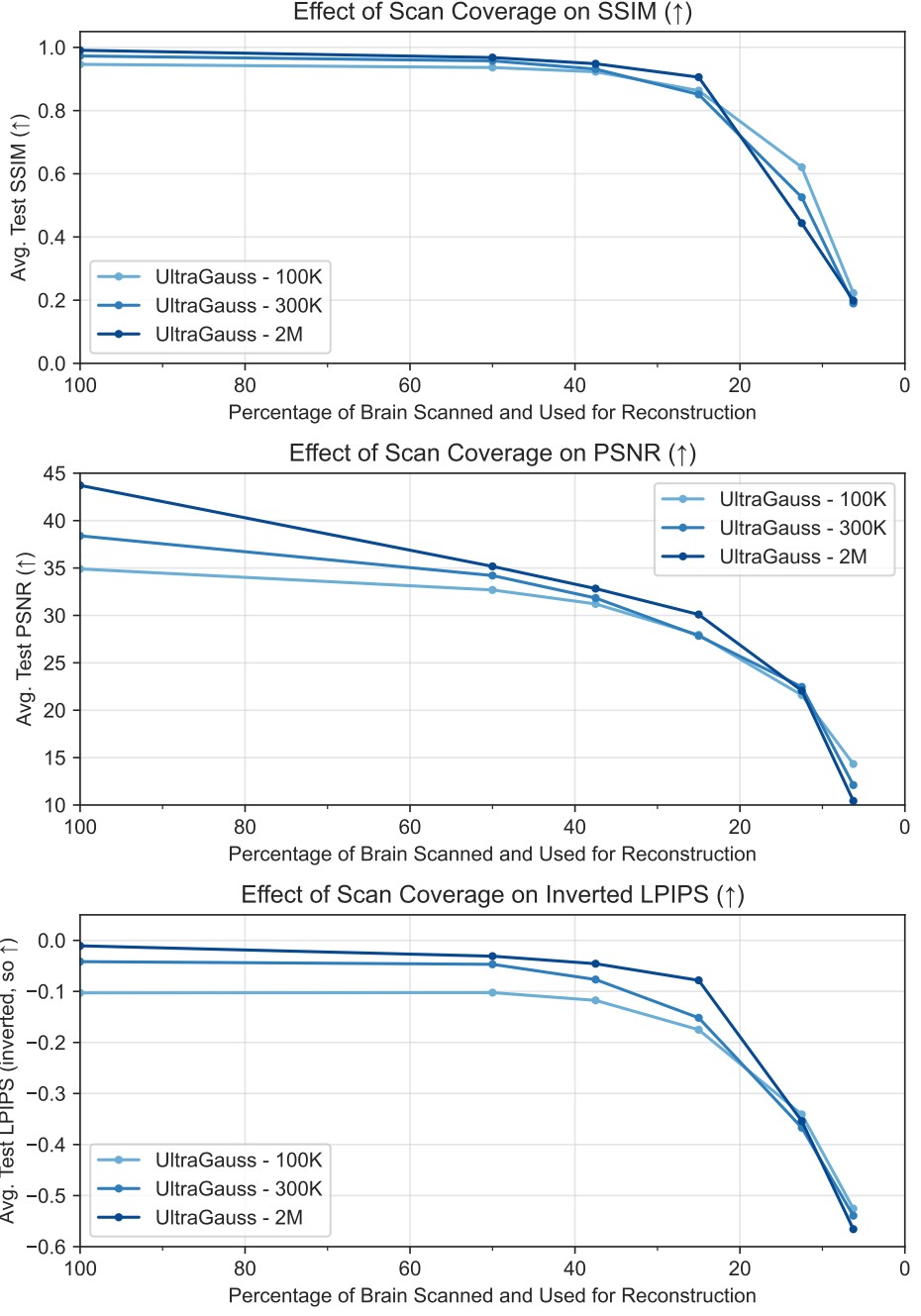

Figure G5: Ablation study on the quality of UltraGauss reconstructions as scan coverage decreases. LPIPS is shown inverted here, so that like SSIM and PSNR, a higher value signifies better accuracy.

### G.6 THE EFFECT OF SCAN SIZE ON RECONSTRUCTION TIME

To confirm our expectation that reconstruction time scales linearly with the number of pixels (i.e. the area of the 2D scans), we conducted an ablation using UltraGauss-300K. Fig. G6 evidently confirms this.

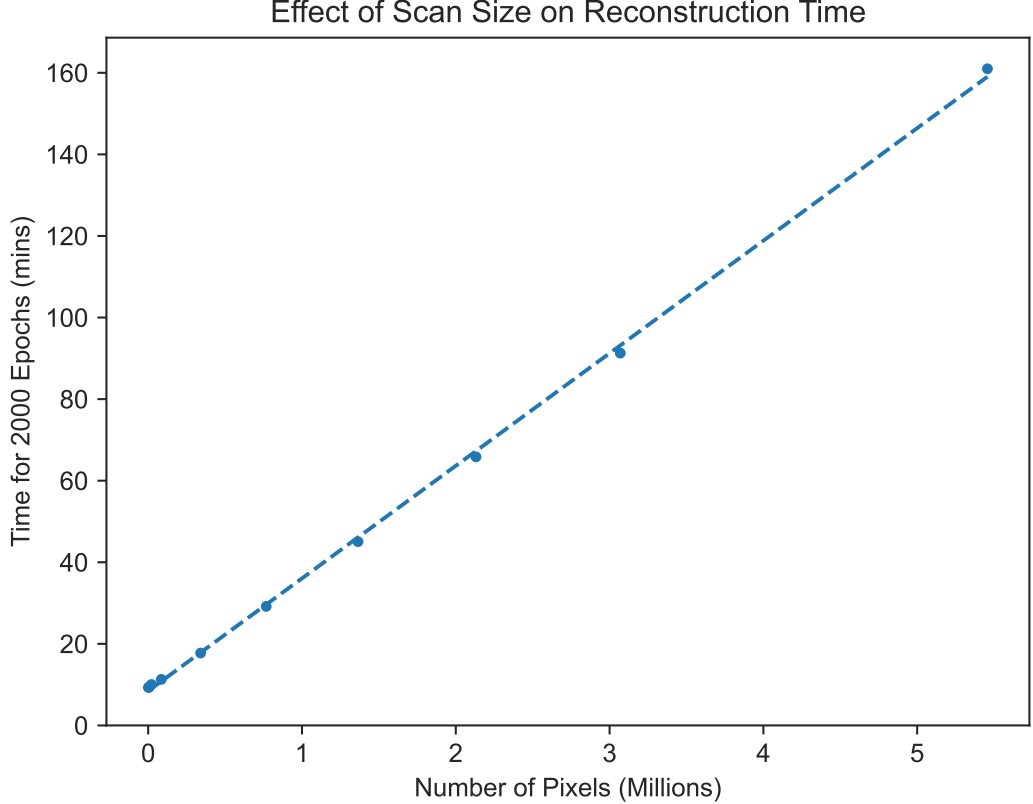

Figure G6: Ablation study on how scans size (and thus the number of pixels), affects reconstruction time of UltraGauss (shown here for UltraGauss with 300,000 initial Gaussians).

