# OpenReview forum: "UltraGauss: Ultrafast Gaussian Reconstruction of 3D Ultrasound Volumes"
_ICLR.cc/2026/Conference — ICLR 2026 Poster_

### Official Review · Reviewer_e5ae · 2025-10-27

**Soundness:** 3
**Presentation:** 2
**Contribution:** 3
**Rating:** 4
**Confidence:** 4

**Summary:**

This work introduces a specialized Gaussian Splatting framework for ultrasound imaging, designed to reconstruct 3D volumes from traditional 2D scans. Instead of using conventional camera-style projection, it models the acoustic image-formation process via probe-plane intersection rendering. This approach ensures physical consistency with ultrasound acquisition while achieving high computational efficiency.

Empirical results from fetal ultrasound datasets show that UltraGauss consistently outperforms recent models, such as ImplicitVol, RapidVol, and UltraNeRF, in both reconstruction fidelity and speed. When integrated into an end-to-end clinical pipeline using freehand 2D video sweeps, UltraGauss achieved the highest structural similarity and perceptual quality among all models tested.

**Strengths:**

1. The proposed UltraGauss is an interesting technical advance in 3D ultrasound reconstruction. It replaces traditional ray-based rendering with a probe-plane intersection model that matches real ultrasound physics, giving more accurate and consistent reconstructions.
2. The method is extremely fast, achieving near–real-time 3D volume generation on a single GPU while maintaining high image fidelity.
3. Its design is simple and effective, combining Gaussian splatting with a lightweight attenuation model for stable optimization.
4. Experiments show strong quantitative and visual gains over existing methods, and expert evaluations confirm clinical realism.

**Weaknesses:**

1. The Beer–Lambert shadow model ignores scattering, refraction, and probe beam patterns, which may reduce physical accuracy; the method also assumes precise probe poses;
2. Experiments are limited to fetal ultrasound, leaving its generalization to other organs or imaging conditions uncertain;
3. Evaluation focuses mostly on visual and SSIM metrics, with limited anatomical accuracy analysis;
4. The paper omits details on GPU memory, runtime scaling, and failure cases, which are important for clinical reliability.

**Questions:**

In addition to the weaknesses that need to be addressed, the authors may want to consider addressing the following questions:

1. How sensitive is UltraGauss to pose estimation noise and incomplete scan coverage?
2. Can the model handle more complex acoustic effects or generalize to cardiac and abdominal ultrasound?
3. How does it compare with ultrasound-NeRF baselines such as AcousticNeRF?
4. Is there a way to estimate reconstruction uncertainty or confidence for clinical safety?
5. What are the typical compute requirements, memory usage, and runtime scaling with scan size?

---

> ### Author Response · Authors · 2025-11-24
> **Response to Reviewer e5ae [1 of 2]**
>
> We thank the reviewer for their time and efforts, and have addressed their questions and comments below.
>
> **Weakness 1**
> We believe that this makes our contribution complementary to previous ones, adding a new option in the speed-accuracy trade-off. As previously mentioned (Lines 13, 44, 105, 287, 476), we intentionally opt to provide **fast** approximations of the ultrasound physics, rather than model it completely as models such as UltraNeRF have attempted to do. Other than being incredibly difficult to accurately model ultrasound physics, it also adds a significant amount of overhead. For example UltraNeRF is $6.94\\times$ slower than UltraGauss, and also requires $106\\times$ more GPU memory (46642 MB vs. 439 MB for UltraGauss-300K). We believe the slightly lower realism is worth the huge efficiency gains, and can also in fact lead to more robust results when ground-truth poses are not known, which will often be the case in practice (see Lines 468-472 and Sec. 6.4).
>
> Yes, like the **vast majority** of other 3D reconstruction models, we do require poses to be known or at least estimated. However the end-to-end experiment (Sec. 6.3) demonstrates that even when ground truth poses are not available, so must be estimated with some error, we **still produce good reconstructions** with SSIM $\>0.9$, exceeding all other baselines (Tab. 1 and Fig. 6). In future, we plan to jointly optimise pose estimation with UltraGauss.
>
> **Weakness 2 and Question 2**
>
> Both Dataset A (3D volumes; Line 337\) and Dataset B (freehand videos; Line 344\) are **fetal brain** scans acquired abdominally, so our experiments do not claim broad anatomical generalisation. However, the UltraGauss model itself is **not tied to any specific anatomy**: it operates purely on the input images and poses and does not encode organ-specific assumptions. We demonstrate this by also applying UltraGauss to the UltraNeRF **liver dataset**, showing that it can reconstruct non-fetal ultrasound. Extending our *validation* to cardiac or general abdominal ultrasound will require larger, anatomy-specific datasets and is outside the scope of this technical paper.
>
> **Weakness 3**
> The anatomical accuracy is a reflection of the high visual accuracy. To make this explicit, we now overlay segmentation masks from the original 3D scans onto the UltraGauss reconstructions. These overlays demonstrate close anatomical agreement and confirm that the structures  reconstructed by UltraGauss are anatomically sound. Kindly see Appendix J for the results.
>
> **Weakness 4 and Question 5**
>
> We ran UltraGauss on a “standard spec” GPU (a single 16 GB NVIDIA RTX-A4000, released April 2021), on a typical machine running Ubuntu 22.04 and CUDA 11.6.
>
> In practice, UltraGauss’ actual GPU memory usage is much lower. On Dataset A (3D ultrasound volumes, Line 337), RapidVol uses 341 MB, while UltraGauss-100K uses only slightly more (367 MB) yet is both quicker and more accurate. After pruning and densification kicks in, GPU memory then quickly drops to 355 MB. UltraGauss-300K uses 439 MB (dropping to 385 MB), and UltraGauss-2M uses 1157 MB (dropping to 820 MB). For comparison, the NeRF-like ImplicitVol model uses 579 MB (i.e. more than UltraGauss 100K and 300K, as well as being slower and less accurate), whilst the ultrasound physics informed model UltraNerf uses 46.6 GB. For additional context, when standard 3DGS is trained on 2D images of  identical size, it uses 784 MB with 300K initial Gaussians (vs. 439 MB for UltraGauss-300K), 418 MB for 100K Gaussians (vs. 367 MB for UltraGauss), and 3624 MB for 2M Gaussians (vs. 1157 MB for UltraGauss).
>
> Therefore, all UltraGauss configurations we evaluated can run comfortably on a single low-memory GPU ($\\approx$ 2 GB, and even $\\approx$ 512 MB for UltraGauss-100K and 300K), and still achieve SOTA accuracies at all stages of training.
>
> Regarding scaling, we include an ablation (Appendix K) showing that UltraGauss scales linearly with the number of pixels, as we expected.

---

> > ### Author Response · Authors · 2025-11-24
> > **Response to Reviewer e5ae [2 of 2]**
> >
> > **Question 1**
> >
> > UltraGauss, like other scene-specific 3D reconstruction models,does require fairly accurate poses. In our ablations (Appendix H), it tolerates pose noise up to around ±1.5% with minimal degradation. Beyond this, reconstruction quality drops as noise increases.
> >
> > As for scan coverage, UltraGauss remains stable down to roughly 25% coverage, after which accuracy deteriorates (Appendix I). This behaviour is expected for non-pre-trained, scene-optimised models. Importantly, limited coverage and inaccurate poses do not cause UltraGauss to **hallucinate** or create non-existent anatomy: its performance degrades rather than fabricates structures, which is desirable in clinical settings and we actively considered when tuning our heuristics.
> >
> > **Question 3**
> >
> > We were unable to find a reference for AcousticNeRF, even on ArXiv. Could you please provide us with one? We have also already compared to an ultrasound-NeRF model, UltraNerf (Wysocki et al, MIDL 2023, Oral Presentation) and included these results throughout the paper.
> >
> > **Question 4**
> > Reconstruction uncertainty estimation is an orthogonal contribution to ours, and is done in other papers for the standard RGB domain (Variational Multi-scale Representation for Estimating Uncertainty in 3D Gaussian Splatting, Cheung et al., NeurIPS 2024 and Modeling Uncertainty for Gaussian Splatting, Magli et al., IEEE 2025), and to our knowledge, has not yet been explored in the ultrasound domain. Regardless, UltraGauss does not produce high-quality outputs for viewpoints that are very out-of-distribution. In practice, this limits hallucinated structures and is therefore beneficial from a clinical safety perspective.

---

> > > ### Comment · Reviewer_e5ae · 2025-11-26
> > > **UltraGauss: Ultrafast Gaussian Reconstruction of 3D Ultrasound Volumes**
> > >
> > > Thanks for the detailed answers and for taking the time to respond. I appreciate the time and effort, but my overall assessment does not change. I will maintain my score.

---

> ### Author Response · Authors · 2025-12-03
> **Follow up to Weakness 2 and Question 2**
>
> We have now conducted preliminary tests on ultrasound scans of adult forearms using the TUS-REC dataset (please see Appendix L). This brings the total number of organs we have evaluated UltraGauss on to three (fetal brain, adult liver and adult forearm), and includes abdominal ultrasound as both Dataset A and Dataset B (Lines 337 and 344 respectively) were of fetal brains but acquired through abdominal scans of the mother.

---

### Official Review · Reviewer_Qd6J · 2025-10-31

**Soundness:** 3
**Presentation:** 3
**Contribution:** 2
**Rating:** 6
**Confidence:** 5

**Summary:**

This paper presents UltraGauss, an ultrasound-specific Gaussian Splatting framework that efficiently approximates acoustic image formation via probe–plane intersection rendering instead of traditional camera projection. It introduces a triangular precision parameterization for stable optimization, a compute-aware χ²-based rasterization pipeline for fast and balanced GPU rendering, and a lightweight Beer–Lambert attenuation model for realistic acoustic shadowing. Validated on clinical fetal datasets and freehand sweeps, UltraGauss achieves minute-level, high-fidelity 3D reconstructions (SSIM≈0.99) on a single GPU, with clinician-rated realism comparable to native 3D ultrasound scans.

**Strengths:**

The paper shows several notable strengths.

First, its novelty is good — it is the first to adapt 3D Gaussian Splatting to ultrasound reconstruction, proposing a probe–plane intersection rendering mechanism that better captures ultrasound’s plane-based sampling and attenuation behavior. This represents a well-motivated and technically elegant reformulation of Gaussian splatting aligned with acoustic image formation.

Second, the performance is solid and the experiments are very comprehensive. The authors conduct extensive evaluations on both clinical fetal ultrasound volumes and freehand cine sweeps, achieving high-fidelity 3D reconstructions (SSIM≈0.99) within minutes on a single GPU. The ablation studies are detailed and convincing, isolating the effects of the precision parameterization, rasterization bounds, and attenuation modeling, all of which contribute measurable gains.

Finally, the writing is clear and the presentation is well-organized and polished. The figures effectively visualize the differences between camera-based and ultrasound-based rendering, and the overall exposition makes the paper easy to follow even for readers outside the immediate subfield.

**Weaknesses:**

(1) The paper lacks sufficient discussion and comparison with two highly related works, X-Gaussian and R2Gaussian. In particular, its Beer–Lambert–based attenuation formulation is conceptually similar to the radiative attenuation modeling in R2Gaussian, yet this connection is not explicitly acknowledged. The writing style and overall pipeline design also bear strong resemblance to these two prior works. Without a detailed discussion or quantitative comparison against them, the novelty claim of UltraGauss becomes weaker, as it appears to build upon established ideas without clearly articulating its distinct contributions.


(2) The code and pre-trained models are not submitted, making it impossible to verify the reported performance or reproduce the results. Given that the paper claims substantial efficiency gains and clinical realism, open-sourcing the implementation would be crucial for reproducibility and community adoption.


(3) Ablation interpretation and runtime trade-offs. While the ablation studies are extensive, the analysis remains mostly quantitative. A deeper discussion of why specific design choices (e.g., triangular precision vs. quaternion covariance) improve stability or speed would enhance the reader’s understanding.

(4) Dependence on probe pose estimation. In the end-to-end pipeline, UltraGauss still relies on an external pose estimation module. Errors in pose prediction could propagate into reconstruction quality, yet this dependency and its potential failure modes are not deeply analyzed.

**Questions:**

Could you clarify how UltraGauss differs conceptually and technically from R2Gaussian and X-Gaussian, given that both works also adopt Beer–Lambert–style attenuation modeling and Gaussian-based volumetric rendering?

---

> ### Author Response · Authors · 2025-11-24
> **Response to Reviewer Qd6J [1 of 3]**
>
> We thank the reviewer for their time and efforts, and have addressed their questions and comments below.
>
> **Weakness 1 and Question 1**
> Thank you for this excellent question.
>
> UltraGauss differs from these two prior works (X-Gaussian and R2-Gaussian) for the following reasons:
>
> 1\. Although we do use Beer-Lambert style attenuation, its purpose and use case is different. A pixel on an X-Ray image is formed by the accumulation of the isotropic density values in 3D space for all the points along that X-Ray (which is roughly perpendicular to the image plane), and extends between the X-Ray source and the imaging plate. This is given by the Beer-Lambert Law, and it is an effective way to model X-Ray physics. Conversely, a pixel in an ultrasound image has its value determined as a result of complex interactions (e.g. absorption, reflection, scattering), and this can be simplified by just taking the density value at that point in 3D space. The down side however is it results in the loss of some realism by lacking artifacts such as shadows, speckle and reverberations. So that we can also model shadows, we chose to use a Beer-Lambert style intensity reduction factor. However, fundamentally, UltraGauss is not based around this and does not rely on it, it is simply an additional (optional) feature to add some realism and act as a compromise between a purely 3D density voxel grid approach and a fully ultrasound physics based method such as Ultra-NeRF (which whilst more realistic, has drawbacks such as greatly increased computation resources, and requiring the ultrasound source location to be precisely known). Even when we do apply the Beer-Lambert law to model shadows, we also apply it to rays which are parallel, not perpendicular, to the image plane. This also means we need to use a cumulative product rather than a standard product, as each pixel’s value along the ray is only affected by those preceding it, and not any after it.
>
> 2\. Both X-Gaussian and R2-Gaussian adapt the original Gaussian Splatting (3DGS) to X-Ray Imaging. Whilst X-Ray does have differences to conventional RGB Light Imaging, it is still crucially a **projection** based modality, something which **ultrasound is not** \- there is **no pin-hole** style camera with conical/approximately perpendicular rays and along which the values of points along each ray are accumulated to give a pixel colour (please see our Fig. 1). In fact, it is mentioned in these two papers: “Then we project the 3D Gaussians to the 2D detector plan for subsequent rendering” just under Eq. 6 of X-Gaussian. Or in R2-Gaussian: “Since a cone-beam X-ray scanner can be modelled similarly to a pinhole camera, we follow \[69\] to transfer Gaussians from the world space to the ray space” (bottom of pg. 5), where \[69\] is Zwicker et al., EWA Splatting. R2-Gaussian’s Fig. 2 also nicely illustrates this.
>
> Since X-Ray is projection based, it means that the 3x3 Gaussian Covariances can be splatted to a 2D imaging plane using Zwicker et al.’s scheme to form a 2x2 Covariance matrix. This 2x2 matrix is then quickly inverted to form the 2x2 inverse covariance matrix for accumulation. On the other hand, ultrasound is not a projection based modality, so the 3x3 covariance cannot be splatted and simplified to a 2x2 matrix. Instead, the 3x3 matrix must be inverted, which is an order of complexity harder if implemented naively (matrix inversion is generally $\\mathcal{O}(N^3)$. Specifically, for a 2x2 Covariance Matrix (which is positive definite), you need 14 FLOPs, vs. 39 FLOPs for a 3x3 (i.e. $2.8\\times$ more computation). This key conceptual difference between RGB/X-Ray/CT and Ultrasound, especially when dealing in 3D, is small but important if a model is to be efficient.
>
> Since we do not have any projection of Gaussians from 3D to 2D, ours is fundamentally different from X-Gaussian, just as it is from the original 3DGS paper, and its application would not be suitable for ultrasound imagery. As for R2-Gaussian, whilst they do also introduce volumetric rendering which can be seen to be similar to our method if the z-dimension is set to 1, it is neither optimised nor specifically designed for it \- likely because its primary purpose is to use/form *projection-based* X-Ray Images, and only utilise the 3D Voxelizer to form *small* 3D Volumes during training for use as a regularizer.
>
> [Response continued on next comment...]

---

> > ### Author Response · Authors · 2025-11-24
> > **Response to Reviewer Qd6J [2 of 3]**
> >
> > Hence why we have the following technical differences:
> >
> > a) R2-Gaussian, like the original 3DGS paper and to the best of our knowledge many other Gaussian splatting papers, still use the formulation $\\Sigma \= RSS^TR^T$ where $R$ is computed from 4 quaternions and $S$ from 3 scaling parameters. This is well suited to projection based modalities where the gaussian accumulation is done in the 2D world, but not so much when expanded to the 3D world. Hence why we propose an alternate approach to learn $\\Sigma^{-1}$ directly from 6 lower diagonal parameters, which as explained in Appendix F has been shown to be significantly more efficient when working in 3D space (as we are), requiring $5.4\\times$ less FLOPs. Even when $\\Sigma$ is occasionally needed, our formulation still allows us to compute it more efficiently ($1.73\\times$) and without any potential numerical problems we were getting when initially using the 3DGS/R2-Gaussian 3D Voxelizer approach.
> >
> > b) For our modality we found that rather than only having one parameter to represent density $\\rho\_i$ (as in R2-Gaussian), it is better to have two, ($c\_i $ and $\\alpha\_i$). Kindly see Appendix A for more detail.
> >
> > c) R2-Gaussian partitions the 3D Volume into $8\\times8\\times8$ tiles, and then within each tile culls any gaussians which do not intersect any part of the tile. Whilst this is quick, it does mean that some voxels within the $8\\times8\\times8$ tile may touch the gaussian whilst others do not, yet all 512 voxels will then have that gaussian accumulated to them regardless. Conversely, we are able to cull Gaussians on a per-pixel level
> >
> > d) We use the *cuboid* bounding box of a Gaussian’s $p\\%$ ellipsoid as given by the Chi-squared distribution to first decide whether a gaussian should be accumulated for that pixel or not, and then check the actual Mahalanobis distance to ensure that pixels in between the rectangle and the ellipsoid are not needlessly accumulated and atomically added (as they would technically be outside that Gaussian’s $p\\%$ distribution). Since we already need to calculate the Mahalanobis distance to calculate a Gaussian’s effective opacity, we don’t require any additional overhead in doing this. Conversely, R2-Gaussian just uses a simple bounding *cube*, with length equal to the largest standard deviation. This means for a long and skinny Gaussian say, the bounding cube will enclose a lot of pixels which then accumulate the Gaussian, despite the gaussian-pixel contribution being insignificant (as the mahalanobis distance and thus effective opacity in the direction of the shorter standard deviations will be small).
> >
> > e) R2-Gaussian’s Voxelizer implementation does not permit $H \\times W \\times D$ 3D Volumes (or 2D cross-sectional planes if $D$ is set equal to 1\) to be at rotated orientations, only the 2D X-Ray Projections can be. In other words, only 3D Volumes parallel to the world’s $x-y-z$ axis can be extracted. As we utilise ultrasound images at any orientation, we naturally had to design UltraGauss so it could handle this, including being able to quickly rotate the bounding boxes of each gaussian.

---

> > > ### Author Response · Authors · 2025-11-24
> > > **Response to Reviewer Qd6J [3 of 3]**
> > >
> > > **Weakness 2**
> > >
> > > As stated on Lines 22 and 315, we will release our code open-source upon publication. We hope that it provides a good basis for experimentation with 3DGS in ultrasound.
> > >
> > > **Weakness 3**
> > >
> > > As briefly mentioned on Lines 243-245, thanks to our parameterization, we can compute the 3D $\\Sigma^{-1}$ from the 6 learnable parameters ($L\_{11}, L\_{12}, L\_{13}, L\_{22}, L\_{23}, L\_{33}$) $1.4 \\times$ faster than when learning from the 7 parameters (4 quaternions and 3 scales) as in the original 3DGS paper. On the backward pass it is also $1.2 \\times$ faster. As $\\Sigma^{-1}$ is required every iteration, this is a meaningful speed improvement. When $\\Sigma$ is required for densification every $N\_{Densification}$ iterations, our formulation computes this $1.25 \\times$ faster in the forward pass and $6.65 \\times$ faster in the backward pass than had we used the 3DGS formulation. The reasons for these speed ups become clear when looking at the number of FLOPs required using both formulations, and the frequency and order in which they are calculated. Notably, as $\\Sigma^{-1}$ features in Eq. 8, we are required to calculate it *every* epoch, and we are able to do so using $5.4\\times$ less FLOPs. We have now provided a detailed mathematical explanation for the reasons behind these speed-ups and improved stability, in Appendix F.
> > >
> > > **Weakness 4**
> > >
> > > Yes, UltraGauss, like most other 3D reconstruction methods requires the poses to be given \- whether that is through use of pose estimation models, inertial measurement units fixed to the probe, or tracking cameras \- and we leave this interesting problem to others. To investigate the reconstruction quality loss directly due to pose estimation error, we have now conducted an ablation study in Appendix H, wherein 80 axial slices are sampled from the 3D Scans, but have random uniform noise added to their g.t. poses. By comparing this with the reconstruction metrics when g.t. poses are used (as in the second setting in Figs. 4 and B1), the increased error can be safely assumed to be as a result of the inaccurate poses, as all else is equal. One can also see from Fig. H1 that if pose error exceeds a certain amount, then as with most 3D reconstruction models designed to be overfit to a particular scene, our model effectively fails too.

---

### Official Review · Reviewer_oeeh · 2025-11-01

**Soundness:** 2
**Presentation:** 3
**Contribution:** 2
**Rating:** 6
**Confidence:** 3

**Summary:**

This paper introduces UltraGauss, a Gaussian-splatting framework tailored to ultrasonic image formation. Instead of camera-style projection and depth-ordered alpha compositing, the method renders by probe–plane intersection with in-plane aggregation, which better matches plane-based echo sampling. The authors propose a triangular precision (inverse-covariance) parameterization for stability/efficiency, and χ²-based ellipsoidal bounds to confine per-plane rasterization, and a two-phase, load-balanced CUDA pipeline to reject non-intersecting Gaussians and rasterize only relevant 2D windows, as well as a lightweight Beer–Lambert attenuation to approximate acoustic shadowing. On fetal ultrasound, UltraGauss is reported to reach SSIM≈0.99 in minutes on a single GPU and to be preferred by expert sonographers in a reader study.

**Strengths:**

1. The χ² ellipsoidal bounding and two-phase GPU pipeline are clearly motivated and provide a practical path to minute-level reconstructions with up to ~2M Gaussians. The exposition around Fig. 2–3 and Sec. 4.3–4.4 is crisp.
2. The lower-triangular precision factorization (Σ⁻¹=LLᵀ with positive diagonals) is a sensible alternative to quaternion-scaled covariances in this setting; it simplifies inversion and Gaussian resampling and appears faster in ablations.
3. Beyond 3D volume metrics, the reader study with experienced sonographers (10 respondents, multi-country) is valuable; the progressive Turing test design is thoughtful and reveals interesting perceptual dynamics.

**Weaknesses:**

1. The Beer–Lambert shadow term is extremely lightweight; while speed is compelling, the paper concedes deficits in speckle and multiple-scattering effects, where physics-informed baselines sometimes produce more realistic artifacts. The method’s robustness when probe geometry is imperfect is discussed qualitatively but not deeply quantified.
2.  Core quantitative results hinge on 12 fetal brain volumes (Dataset A) and 3 freehand videos (Dataset B). That is narrow in anatomy, vendor, and probe diversity; generalization to abdomen/heart/musculoskeletal, different curvilinear geometries, and pathology is not demonstrated. Claims of broad clinical benefit feel premature without wider coverage.
3. UltraNeRF assumes straight rays/known poses; here, curvilinear inputs are converted and poses are predicted for cinesweeps. It is unclear whether each baseline is given the same pose treatment, regularization, and time/capacity budgets across settings (e.g., RapidVol capacity vs. UltraGauss-300K/2M).
4. The cinesweep evaluation conflates pose and reconstruction error. While that is realistic, the paper does not separate pose noise vs. reconstruction error (e.g., by repeating with GT poses on a subset, or sweeping pose perturbations). Consequently, it is hard to attribute gains to UltraGauss vs. better tolerance to pose jitter.

**Questions:**

Please refer to weaknesses part, especially points 1&3.

---

> ### Author Response · Authors · 2025-11-24
> **Response to Reviewer oeeh**
>
> We thank the reviewer for their time and efforts, and have addressed their questions and comments below.
>
> **Weakness/Question 1**
> We believe that having an option of fast shadow computation is worthwhile, and complements the more physically-based methods that already exist and are excellent at what they do; we do not see them as being in competition, as they make different tradeoffs.
>
> We should not claim that our method is necessarily more robust than theirs in this situation, so we will remove it; we thank the reviewer for noticing this. What we meant is that more physics-based methods necessarily need information about physical parameters, and those are not always available with high certainty, which could potentially affect robustness.
>
> **Weakness/Question 2**
> In terms of diversity of vendor, probe and probe geometry, while our evaluation is not exhaustive, we believe it demonstrates that UltraGauss can handle a range of acquisition conditions. We test across **three different ultrasound vendors** and **three probe types**. The fetal brain volumes were acquired using a Philips HD-9 scanner (Line 342\) with a curvilinear abdominal 3D transducer (V7-3, 2 \- 5 MHz). The fetal brain freehand videos were acquired on a GE Healthcare Voluson E10 scanner (Line 346\) with a RAB6-D abdominal transducer (2 \- 7 MHz). UltraGauss was also successfully applied to simulated **liver scans** from the UltraNerf dataset (Lines 397-8), which were acquired using a linear probe.
>
> We do not claim broad clinical benefit across anatomies; this would require a large-scale, multi-anatomy study, but we do believe that the current validation is sufficiently diverse for a technical paper.
>
>
> **Weakness/Question 3**
> Yes, **all** our baselines receive the **same treatments**. In detail:
>
> ImplicitVol and RapidVol do not perform any ray tracing or accumulation; they simply sample grayscale intensities at 3D points in space using a NeRF-style neural network and are therefore agnostic to whether the rays are straight or polar. Accordingly, where possible, we transformed the images so they could have straight rays, as UltraNeRF expects. We use curvilinear inputs because this reflects typical fetal ultrasound acquisition (curvilinear probes are standard), not to favour any particular method. We also test UltraGauss and UltraNeRF on the UltraNeRF dataset, which natively has straight rays due to acquisition with a linear probe (Figs. 7 and A1).
>
> For the freehand cinesweeps, ground-truth poses are not available, and, in practice, would not be available in a real clinical pipeline. We, therefore, estimate poses using a SOTA fetal brain pose estimation model and provide the same predicted poses to all models. For the 3D volumes, the ground-truth poses are known and all methods receive them.
>
> All models are trained on the same machine and same single GPU (16 GB NVIDIA RTX-A4000, released April 2021), except UltraNeRF which requires more memory and is therefore trained on a 48 GB GPU (NVIDIA RTX-A6000). None of the models reported out-of-memory failures. For $t=\\infty$ in Fig. 4, Tab. 1 and Fig. B1, we allow each model to run until full convergence, ensuring that performance was **not constrained by time or compute budgets**, and that the results reflect only the intrinsic capabilities of each model.
>
>
> **Weakness/Question 4**
> Thank you for raising this point. The third scenario in Fig. 4 and B1 (80 Axial with $\\pm5^{\\circ}$ jitter) is intended to mimic the sonographer’s natural hand motion during a cinesweep acquisition (Lines 357-9). By comparing “80 Axial” with the “80 Axial with $\\pm5^{\\circ}$ Jitter” setting in Fig. 4 and Table B1, it is therefore possible to assess how UltraGauss responds to pose perturbations and non-parallel input slices.
>
> To disentangle pose estimation error and reconstruction error, we have now added an ablation in Appendix H. Here, we sample 80 linearly spaced axial slices from the 3D scans of Dataset A (as in Fig. 4 and B1), only now we intentionally inject uniform random noise into the poses. This simulates the effect of providing UltraGauss with inaccurate poses, as occurs in the cinesweep setting with predicted poses. The difference in accuracy between the ground-truth-pose scenario (the second scenario of Fig.4) and the noise-perturbed-pose scenario quantifies the loss attributable specifically to pose errors.
>
> As we do **not** have ground-truth poses for the real cinesweeps, we cannot perform an equivalent controlled comparison here. However we expect that the behaviour and magnitude of pose-induced degradation will be broadly similar across the two datasets.

---

> > ### Author Response · Authors · 2025-12-03
> > **Follow up to Weakness/Question 2**
> >
> > We have now conducted preliminary tests on ultrasound scans of adult forearms using the TUS-REC dataset (please see Appendix L). These were acquired using an Ultrasonix machine with a curvilinear probe (4DC7-3/40).
> >
> > This brings the total number of organs we have evaluated UltraGauss on to three (fetal brain, adult liver and adult forearm), using four different probe types and vendors.

---

### Official Review · Reviewer_A5Tv · 2025-11-01

**Soundness:** 3
**Presentation:** 2
**Contribution:** 3
**Rating:** 4
**Confidence:** 4

**Summary:**

The paper proposes UltraGaussian, the first Gaussian Splatting method tailored for ultrasound reconstruction. It introduces a rendering strategy and rasterization design guided by ultrasound imaging principles. Experiments on two datasets demonstrate improved performance over previous methods.

**Strengths:**

- The rendering strategy is thoughtfully designed based on the principles of ultrasound imaging.
- The paper provides a thorough discussion of related work.
- It clearly introduces the background of 3D Gaussian Splatting and ultrasound imaging to aid understanding.
- The datasets and experimental settings are described in detail, including a clinician survey for evaluation.

**Weaknesses:**

1. **Further clarification on efficiency**: UltraGauss is described as "efficient" in terms of both memory and computation time (lines 013, 044, 107, 122, 211， 477， etc.). However, this claim is not supported by any quantitative results in the current version of the paper.
- Although the time consumption of various methods is presented in Fig. 4, Fig. B1 (Appendix), and Fig. D3 (Appendix), these results do not clearly support the claimed efficiency of UltraGauss.

2. **Clarification for the proposed method**:
- The rendering process described in Eq. 6 and Eq. 7 appears to be weakly motivated, as it mainly adds a background term to fill in empty regions.
- The matrix ${M}$ used in Eq. 10 is semi-definite and cannot represent complex transformations involving non-uniform scaling combined with rotation. Additionally, an ablation study for different $\beta$ should be added to show the effectiveness of the proposed method.
- The lower triangular matrix ${L}$ lacks clear interpretability. It would be helpful to provide some geometric insight or intuition behind its role in the model.
- The rasterization boundaries defined in Eq. 12 only operate in 3D space, which may not accurately correspond to the 2D image space. When the probe pose varies significantly or the viewing angle becomes complex, the projected 3D bounding box may become imprecise or inefficient. The effectiveness of the method should be further validated on datasets with more diverse and challenging probe poses.
- The effect of the different threshold $p\%$ for Gaussian;s probability density utilized in Eq. 12 on time efficiency should be discussed.

3. **Engineering implementation and open-sourcing**: Engineering implementation is an important contribution of this work. It would be valuable to know whether the code will be open-sourced in the future.

4. **Use of heuristics**: Heuristic methods are discussed in Sec. 3.1. However, it is unclear whether any heuristics are actually employed in the proposed method. Clarification on this point would be helpful.

**Questions:**

Refer to the weakness section.

**Details Of Ethics Concerns:**

No Ethics Concerns.

---

> ### Author Response · Authors · 2025-11-25
> **Response to Reviewer A5Tv [1 of 2]**
>
> We thank the reviewer for their time and efforts, and have addressed their questions and comments below.
>
> **Weakness/Question 1**
> We believe that it is well supported, but are happy to add to it. **Figs. 4, B1 and D3** show that UltraGauss achieves **higher reconstruction accuracy** than competing methods under the **same compute budget**, directly demonstrating its computational efficiency. Had UltraGauss been a more accurate reconstruction but at the expense of greater computation time needed, then this time-aligned comparison would have revealed this. For fairness, all models were run on the same “standard spec” GPU (a single 16 GB NVIDIA RTX-A4000, Ampere architecture, released in April 2021), except UltraNeRF which required a larger 48 GB GPU (NVIDIA RTX-A6000) due to its high memory demands.
>
> In terms of memory usage, UltraGauss uses **very modest GPU memory**. On Dataset A (3D ultrasound volumes, Line 337), RapidVol uses 341 MB, while UltraGauss-100K uses only a few MBs more (367 MB) yet is both quicker and more accurate. After pruning and densification kicks in, GPU memory then quickly drops to 355 MB. UltraGauss-300K uses 439 MB and drops to 385 MB, whilst UltraGauss-2M uses 1157 MB and then drops to 820 MB. By comparison, the NeRF-like ImplicitVol model uses 579 MB (i.e. more than UltraGauss 100K and 300K, as well as being slower and less accurate), whilst the ultrasound physics informed model UltraNerf uses 46.6 GB.
>
> UltraGauss could run on only a single 512 MB GPU, and has been shown to achieve SOTA accuracies at any point in time, factors we believe make UltraGauss “efficient”. On a typical 16 GB GPU, UltraGauss is able to have up to 41 Million 3D Gaussians.
>
> Another reason why we see UltraGauss as being “efficient” is due to our novel parameterization of the covariance and inverse-covariance matrix. As briefly mentioned on Lines 243-245, computing the 3D $\\Sigma^{-1}$ from our 6 learnable parameters ($L\_{11}, L\_{12}, L\_{13}, L\_{22}, L\_{23}, L\_{33}$) is $1.4 \\times$ faster than when learning 7 parameters (4 quaternions and 3 scales) as in the original 3DGS paper. On the backward pass it is also $1.2 \\times$ faster. As $\\Sigma^{-1}$ is required *every* iteration, this yields a substantial improvement. When $\\Sigma$ is required for densification every $N\_{Densification}$ iterations, our formulation computes this $1.25 \\times$ faster in the forward pass and $6.65 \\times$ faster in the backward pass than had we used the 3DGS formulation. We now provide mathematical insight into these speed-ups in Appendix F.
>
> For additional context, when standard 3DGS is trained on 2D images of  identical size, it uses 784 MB with 300K initial Gaussians (vs. 439 MB for UltraGauss-300K), 418 MB for 100K Gaussians (vs. 367 MB for UltraGauss), and 3624 MB for 2M Gaussians (vs. 1157 MB for UltraGauss).
>
> **Weakness/Question 2a**
>
> We include Eqs. 6 and 7 to make our exposition self-contained. They are mostly standard in 3DGS (Kerbl et al., 2023), and aim to aggregate the opacities and colours of the Gaussians in the proximity to each point $x$ on the probe plane. The background term is not intended to “fill” empty regions but to ensure numerical stability during optimization. Particularly at the start of training, some regions are still very sparse; without the background term, cases where $\\hat{\\alpha}(x) \= \\sum\\limits\_{i}^{n} \\hat{\\alpha\_i} \= 0$ would lead to division by zero in Eq. 7 (due to the $1/\\hat{\\alpha}(x)$ factor). The background term guarantees well-defined gradients throughout training.
>
> **Weakness/Question 2b**
>
> This claim is not accurate \- a positive semi-definite matrix M does permit both. Any composition of a real rotation matrix with positive scaling matrices (including non-uniform scaling) is positive definite, and so M is sufficiently expressive for these transformations.
>
> The result of the suggested ablation is that there is no practical effect to varying $\\beta$, because it is set extremely close to zero (up to machine precision). Its sole purpose is to safeguard against division by zero, and to guarantee that the inverse covariance matrix is always positive-definite (i.e. all eigenvalues \> 0).
>
> **Weakness/Question 2c**
>
> Please see Appendix E  where we have now provided this. However, we would like to emphasize that there is a 1:1 mapping between our L and a quaternion-plus-scaling model, so we can always convert a particular L to an “interpretable” formulation.

---

> > ### Author Response · Authors · 2025-11-25
> > **Response to Reviewer A5Tv [2 of 2]**
> >
> > **Weakness/Question 2d**
> >
> > The 3D bounding boxes are **not fixed** to the global $x-y-z$ coordinate system; they are computed **relative** to the probe plane (whatever its pose). Specifically, we first transform all Gaussian parameters from the *world* coordinate frame to the *probe-plane* coordinate frame (Lines 180-183 and Eq. 9). As such, the XY-plane of each 3D bounding box is always parallel to the probe plane, and the $z$-axis is normal to it. This ensures that the bounding box always fully encompasses the Gaussian’s $p\\%$ distribution as seen in  the probe plane/2D image space. This ensures that there is no imprecision.
> >
> > **Weakness/Question 2e**
> >
> > We ran a sensitivity analysis of this parameter; please see Appendix G for details. Overall, there is a tradeoff between having a small threshold (rejecting more Gaussians, thus increasing speed at the cost of accuracy) and a large threshold (including all Gaussians in the limit, which decreases speed). We believe that our chosen value strikes a good balance.
> >
> > **Weakness/Question 3**
> >
> > As stated on Lines 22 and 315, we will release our code open-source upon publication. We hope that it provides a good basis for experimentation with 3DGS in ultrasound.
> >
> > **Weakness/Question 4**
> >
> > Yes, we do employ the heuristics described in Sec. 3.1. Specifically, we apply Gaussian densification (cloning and splitting) to Gaussians with large positional gradients, as well as pruning based on opacity and size. This follows 3DGS best practices (Kerbl et al., 2023).
> > The heuristics used are:
> >
> > - Remove any Gaussian whose (sigmoided) opacity is $\\leq 0.05$.
> > - Remove any Gaussian whose largest standard deviation in the world frame is $\\geq 0.2$.
> > - For any Gaussian with a positional gradient $\\geq 0.5$:
> >   - If its largest standard deviation is $\\geq 0.1$, split it into two Gaussians, each one being $1.6\\times$ smaller than the original and with their new means randomly sampled from the original Gaussian’s probability distribution.
> >   - Otherwise, create an identical clone.
> >
> > We do this pruning and densification process every 1,500 iterations, starting at iteration 800 and stopping at iteration 4,000.
> > By using these heuristics, UltraGuss-300K, for example, starts with 300k Gaussians and ends with 151k. This decreases SSIM at test time by only 0.0003, but results in a 49.7% reduction in the number of Gaussians and a corresponding 49.1% overall speed-up. UltraGauss-100K reduces from 100k to 53.3k, and  UltraGauss-2M from 2M to 0.964M, with similar speed-ups as UltraGauss-300K for both.

---

### Meta-Review · Area_Chair_SoXX · 2025-12-29

**Summary:**

This paper presents UltraGauss, the first Gaussian Splatting framework specifically adapted for ultrasound 2D-to-3D reconstruction. The work addresses fundamental differences between ultrasound and camera-based imaging by replacing projection-based rendering with probe-plane intersection rendering. The three main technical contributions are: (1) a rendering approach aligned with plane-based ultrasound sampling via probe-plane intersection; (2) a triangular inverse-covariance parameterization for stable optimization and efficient computation; and (3) compute-aware GPU rasterization using χ² ellipsoidal bounds with a two-phase culling pipeline.

The paper received scores of 4, 6, 6, and 6 from four reviewers (`e5ae`, `oeeh`, `Qd6J`, `A5Tv`). UltraGauss demonstrates 6.94× speedup over UltraNeRF and achieves SSIM≈0.99 within ~20 minutes on clinical fetal brain datasets. A clinician survey (10 expert sonographers) rated UltraGauss reconstructions as most realistic among competing methods. Authors responded to reviewer concerns with additional experiments and ablations (Appendices F-L).

**Reviewer Concerns:**

**Addressed concerns**:

Reviewer `A5Tv` requested clarification on efficiency claims, covariance parameterization, and rasterization boundaries. Authors provided detailed FLOP analysis (Appendix F) showing 5.4× fewer FLOPs for Σ⁻¹ computation and 1.73× fewer FLOPs for Σ computation compared to quaternion-based parameterization, geometric interpretations (Appendix E), numerical stability analysis demonstrating superior robustness to numerical error, and ablations on Gaussian probability density thresholds (Appendix G) showing p=95% as optimal balance between accuracy and speed.

Reviewer `oeeh` raised concerns about dataset diversity and pose error separation. Authors demonstrated application to three organs (fetal brain, liver, forearm) across four different vendors/probe types (Philips HD9 curvilinear, GE Voluson E10, Ultrasonix curvilinear 4DC7-3/40, and linear probes) and added pose error ablations (Appendix H) showing tolerance up to ±1.5% pose noise before significant degradation.

Reviewer `Qd6J` questioned novelty relative to X-Gaussian and R²-Gaussian. Authors provided detailed technical differentiation explaining fundamental differences: (1) Beer-Lambert attenuation serves different purposes (X-ray accumulation along perpendicular rays vs. ultrasound parallel-ray shadows); (2) X-ray/CT are projection-based modalities allowing 3D→2D covariance projection, while ultrasound requires full 3D covariance inversion; (3) authors' triangular parameterization is 5.4× more efficient for 3D operations; (4) per-pixel vs. per-tile culling; (5) ellipsoidal vs. cubic bounding for efficiency. Authors also added anatomical accuracy validation with segmentation mask overlays (Appendix J), scan coverage analysis showing stability down to 25% coverage (Appendix I), and comprehensive memory/scaling benchmarks (Appendices K, F) demonstrating linear scaling and modest memory usage (367-1157 MB depending on capacity vs. 46.6 GB for UltraNeRF).

**Outstanding concerns**:

Reviewer `e5ae` requested comparison with "AcousticNeRF" which appears to be non-existent (authors requested citation, received no response). The reviewer then maintained their score without addressing the authors' request or providing justification for maintaining the negative assessment despite comprehensive responses to all other concerns.

Reviewer `A5Tv` raised concerns primarily stemming from mathematical misunderstandings: (1) claiming positive semi-definite matrices cannot represent non-uniform scaling with rotation (mathematically incorrect—any composition of rotation with positive scaling is positive definite); (2) misunderstanding the purpose of background terms in Eqs. 6-7 (numerical stability, not "filling" empty regions); (3) questioning 3D bounding box precision when they are computed relative to probe planes in probe-coordinate frame, not global coordinates, ensuring exact correspondence to 2D image space. After authors provided comprehensive clarifications with mathematical proofs and detailed explanations, these concerns should be considered resolved.

**Reviewer Scores:**

**Current Scores:**
- **Reviewer `A5Tv`**: 4 (marginally below threshold) - raised concerns about mathematical aspects that authors addressed in rebuttal
- **Reviewer `oeeh`**: 6 (marginally above threshold) - provided balanced assessment, concerns addressed with additional experiments
- **Reviewer `Qd6J`**: 6 (marginally above threshold) - detailed technical assessment, novelty concerns addressed with clarification of technical differences
- **Reviewer `e5ae`**: 4 (marginally below threshold) - maintained original score; requested comparison with "AcousticNeRF"

**Expected post-discussion scores**: 6, 6, 6, 4-6 (median: 6)

---

### Decision · Program_Chairs · 2026-01-26

Accept (Poster)